# Distance-decay reveals contrasting effects of land-use types on arthropod community homogenisation

Orsi Decker [1,2] ✉, Johannes Uhler[2], Sarah Redlich [3], Anne Chao[4], Mareike Kortmann [2], Ingolf Steffan-Dewenter [3], Cynthia Tobisch [3], Jörg Ewald [5], Jana Englmeier[2], Ute Fricke [3], Cristina Ganuza [3], Maria Haensel [6], Vedran Bozicevic[7], Jérôme Morinière[7], Jie Zhang [3] & Jörg Müller [1,2] ✉

Homogenisation caused by intensive land-use is one of the drivers of global biodiversity decline. However, the contribution of land-use intensity to insect diversity loss is still largely untested. Therefore, we compare the rate of species homogenisation of ~12.000 arthropod species (using Barcode Index Numbers) from 450 families, using distance-decay relationships. We study communities along an increasing local land-use intensity gradient from forests to managed grasslands, to arable lands and to settlements situated within near-natural, agricultural, and urban regional landscapes. Our approach for incidence data under consideration of incomplete samples identifies that grasslands harbour the most homogenous communities after taking frequency and species traits into account. In contrast, the most modified land-use types, settlements, and arable lands do not differ from forests, and showed the most heterogeneous communities between locations. Large- and low-mobility species communities are the most heterogeneous in space, but patterns are dependent on local land-use. Regional landscapes modify the community response to local land-use types: near-natural landscapes reduce homogenisation, while agricultural landscapes increase homogenisation. Based on our findings we recommend enhanced conservation efforts particularly in managed grasslands to reverse homogenisation, while settlements and arable lands could be considered more in arthropod community heterogenisation.

Human activities shape the diversity and composition of species assemblages both locally and at a regional scale. Excessive habitat modification drives species composition, favouring specific sets of species, dramatically changing their composition in natural habitats[1,2].

Habitat modification includes intense land-use, which causes both biotic homogenisation and heterogenisation[3,4], when communities can become more or less similar across space[5,6]. This process can be measured via changes in β-diversity, describing community-scale

[1]Bavarian Forest National Park, Grafenau, Germany. [2] Conservation Biology and Forest Ecology, Biocenter, University of Würzburg, Rauhenebrach, Germany. [3]Department of Animal Ecology and Tropical Biology, Biocenter, Julius-Maximilians-Universität Würzburg, Würzburg, Germany. [4]Institute of Statistics, National Tsing Hua University, Hsin-Chu, Taiwan, ROC. [5]Institute for Ecology and Landscape, University of Applied Sciences Weihenstephan-Triesdorf, Freising, Germany. [6]Professorship of Ecological Services, Bayreuth Center of Ecology and Environmental Research (BayCEER), University of Bayreuth, Bayreuth, Germany. [7]Advanced Identification Methods GmbH (AIM), Leipzig, Germany. ✉e-mail: Orsolya.Decker@npv-bw.bayern.de; joerg.mueller@uni-wuerzburg.de

patterns driven either by the environment or stochastic factors[7] (see Supplementary Note 1 for used definitions). Socolar et al.[8] hypothesised a bell-shaped response of β-diversity, with increasing values at moderate land-use intensity (heterogenisation), but a sharp decline with high land-use intensity (homogenisation). Such a homogenisation effect was demonstrated for increasing mowing intensity in managed grasslands[9]. Although increasing land-use intensity is predicted to drastically increase biotic homogenisation[6,10], global studies showed no clear net diversity loss in the Anthropocene[11], but strong taxa-dependent trends[3,12].

An improved understanding of the influence of land-use on β-diversity of arthropods is critical not only to maintain biodiversity, but also from a functional perspective[13]. Homogenisation often degrades ecosystem functions via losing sets of species from the community[14–16], and arthropods, especially insects, play crucial roles in ecosystem functioning[17–20]. Despite their high diversity[21], there is still an ongoing debate on insect declines due to human activities[22–24]. An open research question is about the impact of land-use on insect whole-of-community responses. Our dataset contributes to the understanding of this question by using all-arthropod data and discovers broad community-level patterns in response to land-use.

Distance-decay of pairwise similarity (hereafter 'distance-decay') can quantify the homogenisation effect on species communities over large scales and between many localities[25,26]. Distance-decay patterns are based on two assumptions: (1) areas closer to each other exhibit similar environments, therefore species composition should be more similar (more homogenous between locations); and (2) there is a physical limit to species dispersal distances. Steep distance-decay slopes or strong distance-decay represent less similar communities (hereafter 'heterogeneous communities'), while flat slopes, or weak distance-decay, indicate more similar communities (hereafter 'homogeneous communitiess') between locations. While global distance-decay studies contribute to the understanding of community responses to environmental changes[27], using intercontinental datasets could be problematic. It is widely shown that, besides spatial scale[28–30], distance-decay also depends on study extent and design[31]. Regional and local diversity measures are used in conservation planning[8], but study designs that offer available data at the appropriate scale are often lacking[32]. Moreover, community change patterns are more likely to be related to complex environmental and climatic differences when data is obtained across ecoregions, covering great geographic distances[10,27,33]. In contrast, management-relevant studies seek to find similar ecosystems, which allows for comparisons of habitats shaped by human impacts instead of large-scale environmental drivers. Comparative studies between land-use types, but within similar ecosystems, are relatively scarce apart from some exceptions[34–37].

To be able to provide a comparative study between multiple land-use types, we consider the inherent incompleteness of insects samples, and apply a framework of community similarity along the Hill numbers [38] standardised by sample coverage[39,40]. Rare and infrequent species often drive observed diversity patterns; therefore, we account for changes in communities focusing on infrequent, frequent and highly frequent species separately[41]. These categories differ from abundance or count-based classifications of species but reflect the incidence-based frequencies among the processed sample units (collection times and sample fractions, see 'Methods'). Species data are then processed with a deterministic approach, and frequency classes are identified based on frequencies within samples a posteriori, which does not reflect the geographic distribution or conservation status of a given species. Given that the distance-decay relationship is a form of β-diversity, it is based on α- and γ-diversity; therefore, this metric strongly depends on sample size, sampling effort, and sample coverage[42] (for more details of α-, β- and γ-diversity metrics, see Supplementary Note 1). As a result, when estimating diversity and assemblage similarity metrics, it is important for assemblages (samples) to be statistically comparable across all study sites, which requires a standardised sample coverage-based analysis[43]. β-diversity values can be falsely modified when using unstandardised datasets: incomplete samples increase β-diversity values artificially[44]. Therefore, we adapt coverage-standardised indices of β-diversity to incidence data.

Here, we compare four land-use types of Central Europe along a gradient of land-use intensity in forests, managed grasslands, arable lands and settlements at the scale of a large federal state—a scale most relevant for political decisions. The design provides a standardised dataset where biogeographic patterns play little role in distance-decay relationships. Instead, arthropod community patterns are driven by land-use intensity within the same ecoregion[45]. Besides the design, another strength of the study is the inclusion of most recovered terrestrial arthropod taxa, which provides a better understanding of whole-of-community responses[46]. Single-taxonomic group approaches to distance-decay are highly valuable[47,48], but are regularly biased towards specific characteristics of the studied group when comparing different environments. We use arthropod assemblages of ~12k species units recovered by Barcode Index Numbers (BIN species, see 'Methods') from 450 families collected with Malaise traps at 179 plots over an area of 70.500 km². We study distance-decay at two scales: local and regional. 'Local land-use' is considered the immediate surrounding of arthropod traps (0.5 ha), which impacts arthropod communities the most[49]. 'Regional landscape' describes the broad area (5.8 × 5.8 km quadrant), not only the realised habitat of an arthropod. First, we test distance-decay patterns at the local land-use scale, because arthropod assemblage variations are best explained by local habitat characteristics[50]. Then, we assess the outcome of these local subsets in three types of regional landscapes, which could modify local processes[51–53].

As all our habitats are at least moderately managed, we hypothesised a continuous decline of β-diversity[5,8], resulting in homogeneous communities over spatial distance—measured by distance-decay—with increasing land-use intensity. Distance-decay is expected to be the strongest in forests, meaning that communities are the most heterogeneous between locations. Then, distance-decay is expected to weaken, meaning that communities become homogeneous between locations with increasing land-use intensity: in the order of managed grasslands, arable lands and settlements (for exact definitions, see Supplementary Note 1). Further, the regional landscape could have add-on effects[54]. We expected stronger distance-decay, resulting in less homogenisation impact of the four local land-use types when situated within near-natural regional landscapes; while distance-decay is expected to be weaker, resulting in a more pronounced homogenisation effect in local land-use types when situated within agricultural and urban regional landscapes.

Besides homogenising communities[9], high land-use intensity creates environmental filters for organisms, affecting communities including species with specific traits[55,56]. We hypothesise that communities including large-bodied and low-mobility species to be less similar between locations (strong distance-decay), while small and high-mobility arthropods to occur more evenly across space (weak distance-decay). This pattern should be driven by the fact that intensely managed habitats select for these traits[57,58]. Large-bodied arthropods require bigger patches of optimal habitat[59], and often favour low disturbance regimes[60,61], most likely due to the limited ability to escape.

The current work extends the study of Gossner et al.[9] from grasslands to other land-use types, which showed increasingly similar arthropod communities with increasing grassland management intensity. Similarly, we expect increasing arthropod community homogenisation along the increasing land-use gradient: from forest to managed grassland, to arable land and settlement. However, we did not find evidence for species homogenisation of communities along the studied gradient. Instead, species communities were the most homogeneous in a less intensely managed land-use type, in grasslands.

## Results

The sampling resulted in 1432 samples (179 study plots with 8 collection times), which gained 11387 individual arthropod taxonomic clusters using the Barcode Index Number method (BIN species) via metabarcoding analysis. Of all arthropod BIN species, 42% could be assigned to species level in 38 orders (Supplementary Fig. 1). Sample coverage was fluctuating between 0.6 and 0.9 for local land-use types, and between 0.5 and 0.9 for arthropod trait categories (Supplementary Fig. 2). For land-use types, we standardised sample coverage to 0.8, and for arthropod traits, to 0.7 in subsequent analyses. Samples below the chosen coverage were extrapolated, and rarified when the sample coverage was above the chosen value. Number of species was highest in the forest systems and lowest in arable lands (Supplementary Fig. 3). Based on standardised species matrices, we produced 21 community similarity matrices (using all data, then subsets for trait data) to run the distance-decay analysis (Fig. 1) expecting the strongest distance-decay in forests, which expectation was not supported within our study design (Fig. 2).

### Land-use types

Managed grasslands showed the weakest distance-decay consistently among land-use types (Fig. 3a). Communities focusing on infrequent species ($q = 0$) were significantly the most homogeneous in managed grasslands, while distance-decay in forests, arable lands and settlements were similar to each other. Then, when focusing on frequent species ($q = 1$), distance-decay within forests and managed grasslands did not differ, but it was still significantly weaker on grasslands than on arable lands and settlements. In communities focusing on highly frequent species ($q = 2$), arable lands had the strongest distance-decay, not different from settlements, but significantly stronger than forests and managed grasslands (Fig. 3b, Supplementary Table 1).

When considering the regional scale, near-natural regional landscapes increased the strength of distance-decay consistently on settlements, arable lands and grasslands, but only had a small impact on forests. Agricultural regional landscapes largely weakened distance-decay in forests and managed grasslands but increased the strength of distance-decay on settlements and arable lands. Urban regional landscape strengthened distance-decay on managed grasslands and forests regardless of species frequency. Urban regional landscape weakened distance-decay on settlements and arable lands, but only in the case of communities focusing on infrequent and frequent species. In communities focusing on highly frequent species, the urban regional landscape slightly strengthened distance-decay on settlements and arable lands (Fig. 4).

### Arthropod traits

On average, forests harboured the largest arthropods significantly ($6.21 \pm 0.05$ mm), but not differing from those in arable lands (Fig. 5a). The smallest species occurred in settlements ($5.85 \pm 0.07$ mm), but not significantly different from those in grasslands and arable lands. Species in arable lands had the highest mobility score ($2.72 \pm 0.003$), significantly higher than those in forests and grasslands; while the other three land-use types were similar in their mobilities (Fig. 5b, Supplementary Table 2).

Body size did not significantly influence distance-decay relationships in most land-use types. Significant differences were only detected in communities focusing on highly frequent species ($q = 2$) in forests, arable lands and settlements. In forests, small arthropods had a stronger distance-decay than medium arthropods. Arthropod body size did not impact distance-decay in grasslands. In arable lands, medium arthropods significantly showed the weakest distance-decay. In settlements, large arthropods had a stronger distance-decay than small ones. (Fig. 6, Supplementary Table 3).

Arthropod communities with different mobility categories exhibited different distance-decay relationships. In forests, high-mobility species had weaker distance-decay than low- and intermediate-mobility species in communities focusing on infrequent species ($q = 0$). In the case of communities focusing on frequent and highly frequent species ($q = 1, 2$), low-mobility species were only significantly different from intermediate-mobility species. In managed grasslands, low-mobility species had significantly stronger distance-decay ($q = 0, 1$), except in communities focusing on highly frequent species ($q = 2$). In arable lands, high-mobility species significantly showed the weakest distance-decay ($q = 0, 1$), but this significance disappeared in communities focusing on highly frequent species ($q = 2$). In settlements, distance-decay was not different among species with different mobility categories (Fig. 7, Supplementary Table 4).

## Discussion

The current work extends the study of Gossner et al.[9] from grasslands to other land-uses, which showed increasingly similar arthropod communities with increasing grassland management intensity. However, we did not find evidence for species homogenisation of communities along the land-use intensity gradient (forest – managed grassland – arable land – settlement). Instead, grasslands harboured the most similar arthropod species over spatial distances, and the two most modified land-use types, arable lands and settlements, exhibited the most heterogeneous arthropod communities between locations.

The most natural local land-use type did not have the strongest distance-decay as expected, meaning that forests did not harbour largely different sets of species between locations. The arthropod communities with higher-than-expected homogeneity in forests are likely driven by current and historic Central European forest management. The mostly uniform management practices lead to even-aged stands with mostly closed canopies and lacking old-growth tree patches[62–64]. As a consequence, even though forests had the highest α-diversity (meaning that this land-use type is the most locally diverse in arthropod species[65]), the created analogous habitats in forests lead to communities consisting of similar species between forest patches[66]. Managed grasslands also showed the consequences of uniform practices. While traditionally managed grasslands are the most species-rich habitat types for vascular plants and insects in Europe[67,68], low-input methods are widely replaced by more intensive management practices[69,70]. The high nitrogen-phosphorus-potassium fertiliser inputs[71,72], frequent mowing and intensive grazing[9,73] result in simplified and floristically homogeneous grasslands[74]. This shapes insect communities to be adapted to uniformly managed grasslands, lowering diversity and community differences between locations[53,75]. In contrast to grasslands, intensively modified arable lands and settlements harboured the most different set of arthropods between locations. Although agricultural practices change the local environment for most species[76–78], these systems can offer a diverse habitat mosaic in time and space. A series of arable lands in a given area can be composed of a variety of different crop plants, which makes land characteristics highly diverse within a small area[79]. The diversity of crops creates a locally unique set of arthropod communities associated with the different grown plants. Indeed, crop field size in Bavaria, where the study was conducted, can be described as small- to medium-scale (~1.6 ha arable field size[80]), compared to the German average (~5 ha[81]). In accordance with recent studies, our results also confirm the importance of high crop heterogeneity and small crop field size. This can be beneficial to biodiversity, even to a greater extent than natural land cover[82,83]. However, Uhler et al.[62] found the lowest insect α-diversity in arable lands, where we found the strongest distance-decay. This pattern could be a result of increased habitat heterogeneity between the compared patches of arable land, combined with decreased average species occupancy. For example, a couple of species could specialise on a given crop in one arable land, while on another arable land, a different plant is grown with a different set of highly specialised (or pest) species. This way, only a few number of

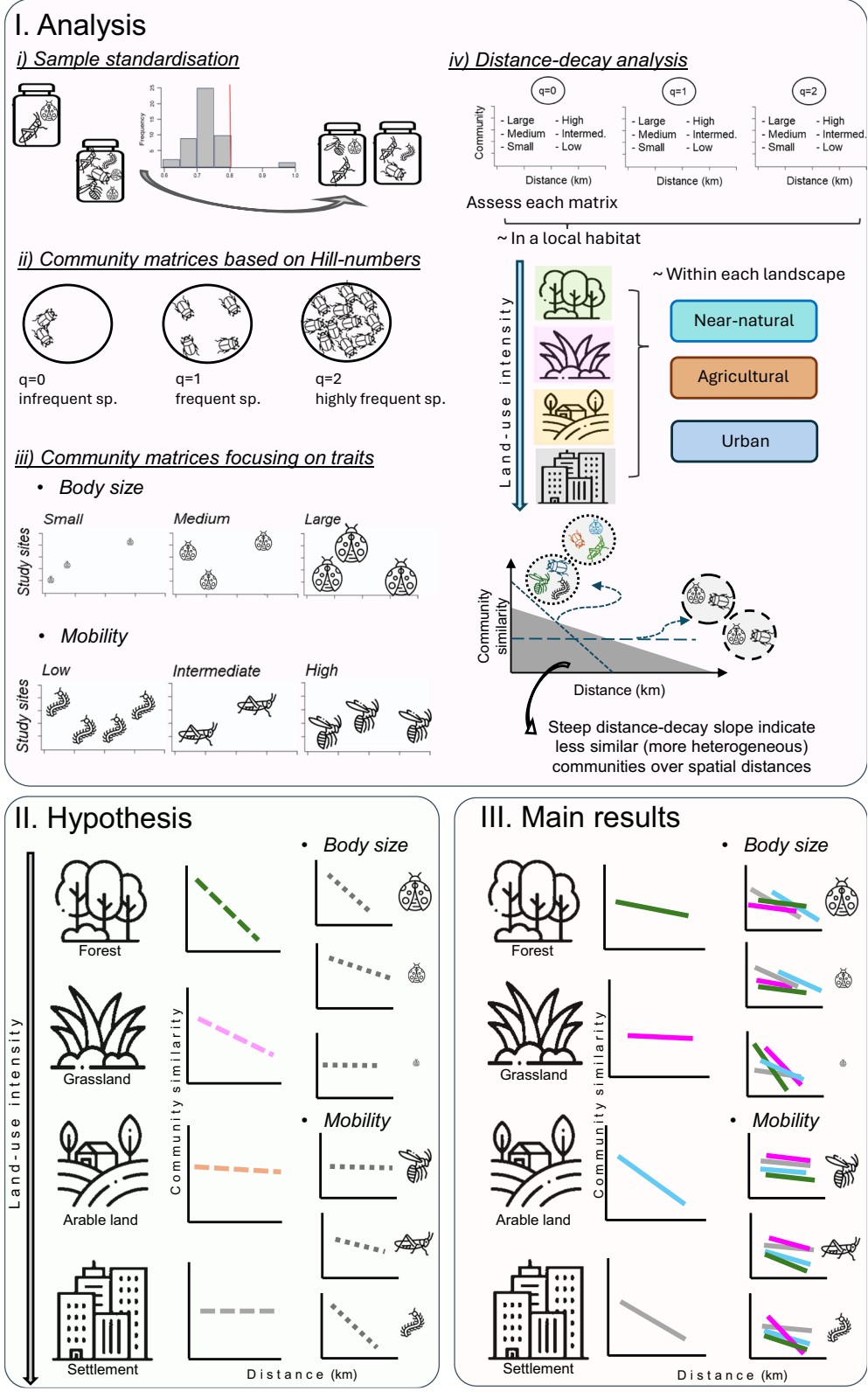

**Fig. 1 | Summary figure showing the analysis, hypothesis and main results of the study.** In our analysis (I), we first standardised our arthropod dataset based on 179 study plots and ~12k arthropod BIN species according to sample coverage, then calculated community metrics along the Hill numbers for all arthropods, and for arthropod subsets based on body size and mobility category. Using this standardised dataset, we analysed distance-decay relationships along a land-use intensity gradient in different landscape types. Based on the homogenisation hypothesis by intense land-use (II), communities should be more similar with intense local land-

use (in settlements = grey and arable land = golden), than in areas with less intense land-use (in managed grasslands = purple and forest = green). However, we found (III) that the most intensively used lands (settlement and arable land) heterogenised arthropod communities when compared to forests and managed grasslands. Body size and mobility did not have a uniform impact on community responses to land-use but depended on rarity and land-use type. Icons designed by afif fudin, Nadiinko, iconfield, Nikita Golubev, Mayor Icons and Freepik on Freepik.

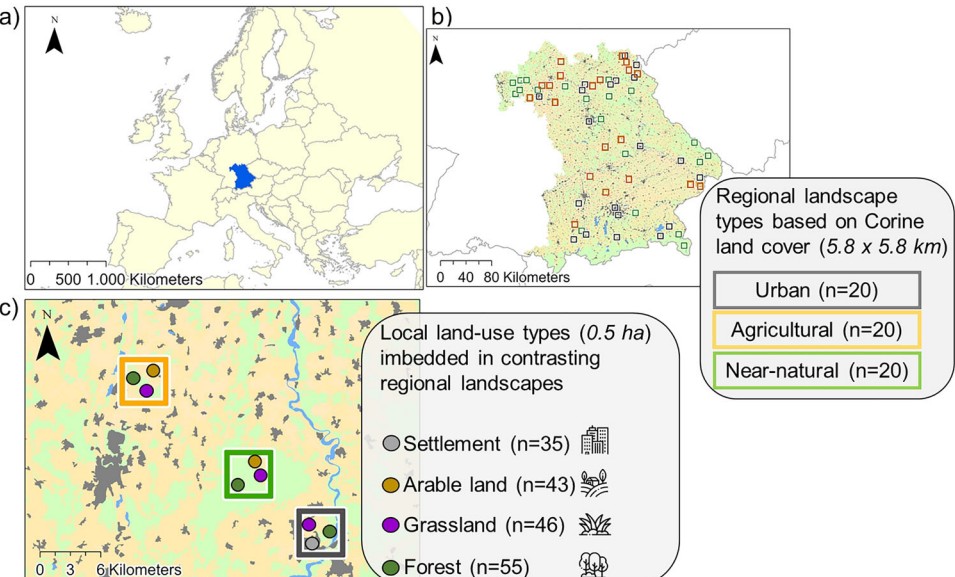

**Fig. 2 | The figure shows the study area and design. a** The map is showing the extent of the study in Southern Germany, Bavaria, where **b** 60 regions were chosen with three landscape types: urban (grey), agricultural (yellow) and near-natural (light green). Within each region, further study plots (**c**) were established based on the local land-use types. In total, 179 study plots were established in four local land-use types: forest (forest green), managed grassland (purple), arable land (golden) and settlement (dark grey) surrounded by contrasting regional landscapes. Icons designed by Mayor Icons and Freepik on Freepik.

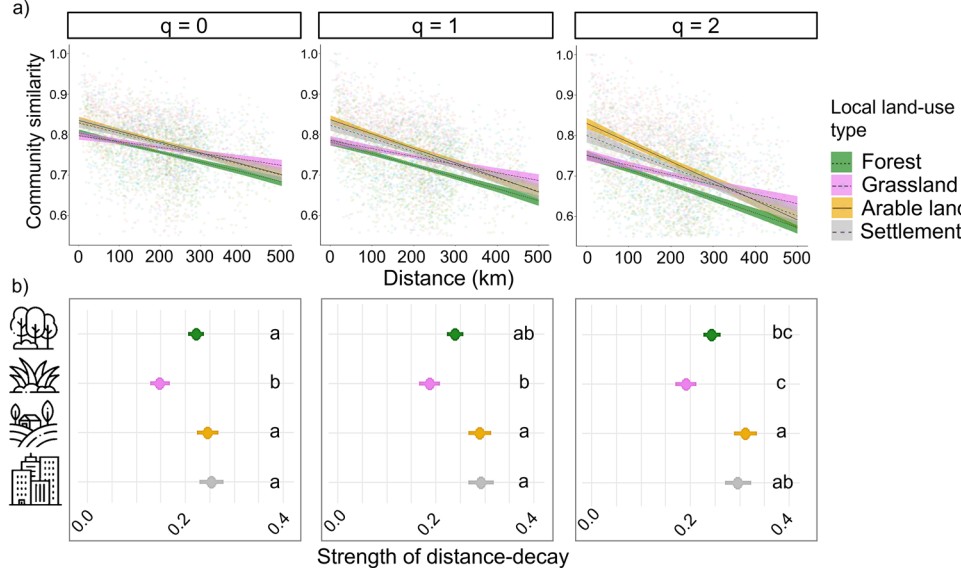

**Fig. 3 | The figure shows distance-decay relationships and slopes in the studied land-use types. a** The graph shows distance-decay slopes (mean ± SE bands) derived from linear regressions in the four local land-use types: forest, $n = 55$ (forest green), grassland, $n = 46$ (purple), arable land, $n = 43$ (golden) and settlement, $n = 35$ (grey) according to Hill numbers. **b** The strength of distance-decay is shown by the absolute value of mean distance-decay slope steepness ± SE multiplied by 1000 and are shown over the Hill gradient: infrequent species: $q = 0$; frequent species: $q = 1$; and highly frequent species: $q = 2$ from left to right panel in the four local land-use types: forest, $n = 55$ (forest green), grassland, $n = 46$ (purple), arable land, $n = 43$ (golden) and settlement, $n = 35$ (grey) according to Hill numbers. Significant differences are shown with different letters according to the SIMBA analysis. Icons designed by Mayor Icons and Freepik on Freepik.

high-occupancy species benefit from each arable land patch, decreasing α-, but increasing β-diversity[3]. Besides crop heterogeneity, differences in agricultural practices could create complexity, which selects for different species sets at different arable lands. For example, different types of intercrop ground cover will change the species which could overwinter at a given arable land; or the method of fertilisation (timing, type of organic input, etc.) can also lead to different species compositions[84]. Differences in practices are also present in settlements, but at a smaller scale: gardens in settlements have a huge range of grown plants and types of garden maintenance. Thus, even though urban expansion is a major cause of habitat loss for many species[85,86], increasing attention is given to the beneficial impacts of dense habitat mosaics for species in settlements[87–89]. Similarly to our finding for β-diversity, Uhler et al.[65] reported high species richness of arthropods in settlements—despite low biomass—especially within near-natural regions.

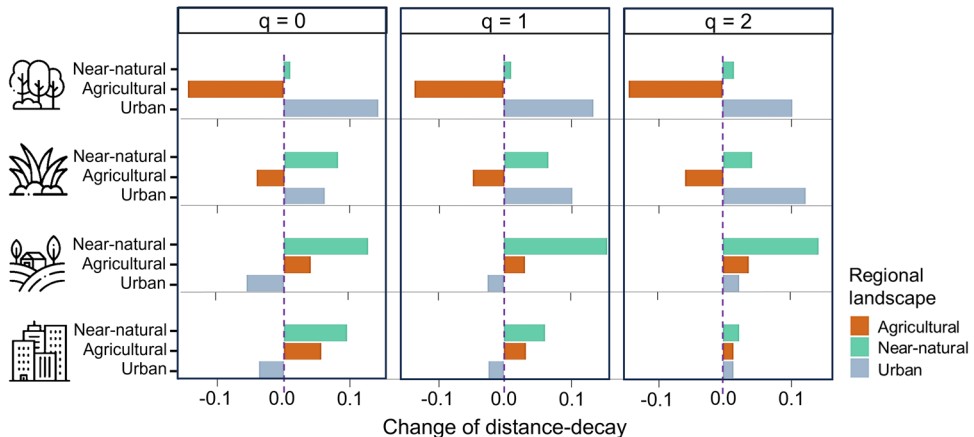

**Fig. 4 | The figure shows the impact of the surrounding landscape on local distance-decay relationships.** The values were calculated by using the difference between the overall distance-decay curves of local land-use types and distance-decay curves considering specific surrounding regions: near-natural (moss green), agricultural (brown) or urban (sky blue). Panels show communities over the Hill numbers: infrequent species: $q = 0$; frequent species: $q = 1$; and highly frequent species: $q = 2$. Bands in the positive area mean less community homogenisation in a given region compared to all samples, and bands in the negative area mean further homogenisation in a given region. Icons designed by Mayor Icons and Freepik on Freepik.

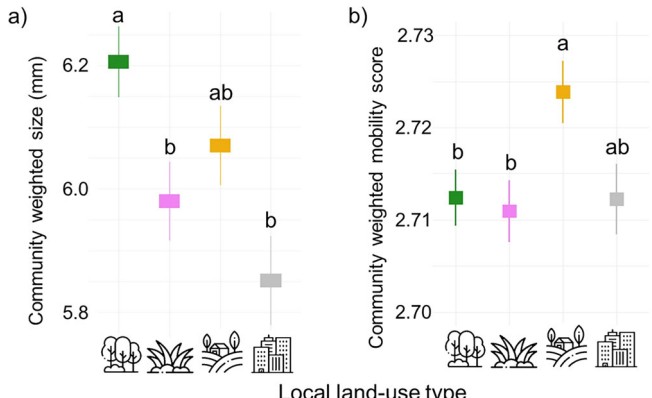

**Fig. 5 | The figure illustrates arthropod community trait values in the studied land-use types.** Graph **a** shows community-weighted body size means ± SE, and graph **b** shows mobility score means ± SE in the four local land-use types: forest, $n = 55$ (forest green), grassland, $n = 46$ (purple), arable land, $n = 43$ (golden) and settlement, $n = 35$ (grey) according to Hill numbers. Body size classes are 0–2.7 mm (small); 2.8–7 mm (intermediate); and bigger than 7 mm (large). Mobility score indicates 1 = low, 2 = medium and 3 = high. Significant differences were tested using analysis of variance with Tukey's correction for multiple comparisons, and significant differences ($p < 0.05$) are shown with different letters. Icons designed by Mayor Icons and Freepik on Freepik.

At the regional scale, near-natural landscapes had a clear mitigating effect, while agricultural landscapes had a worsening homogenisation impact. Near-natural regional landscapes could act as insurance for wildlife[90], according to the 'landscape-moderated insurance hypothesis'[91]. Natural patches, such as semi-natural forests, provide a more stable environment compared to managed grassland or arable land which have frequent rotations between harvest and growth periods. When the more intense land-use types are surrounded by near-natural landscapes, species can move between these areas. Species can escape disturbance periods, such as harvesting on arable lands and mowing on grasslands, to the surrounding near-natural areas and recolonise from there[92]. This process increases species pools locally, therefore can increase community heterogeneity between locations[93]. At the same time, being surrounded by agricultural areas further increased species homogenisation at low-intensity land-use types: forests and grasslands. This result shows that broad-scale agriculture has a detrimental impact on nearby communities in natural and semi-natural local habitats[53,94]. On the other hand, likely the most modified landscape, the urban regional landscape increased community heterogeneity on the least modified local land-use types (forest and grassland). This heterogenising effect could imply that forests are isolated by urban environments[95]. This way, arthropod species became limited in their dispersal, creating unique communities within each individual forest surrounded by urban landscapes. However, because this strong isolating impact by urban regions was not apparent in any other local land-use types in our study, the more likely explanation is that urban regions include highly diverse structures, which further heterogenise forest communities between locations[96]. Urban features add diverse hospitable habitats for otherwise forest-inhabiting arthropods, such as small hobby farms, private gardens, riparian corridors and even small remnant vegetation[97,98]. Such contrasting habitat types could act as complementary or supplementary environments, explained by the 'cross-habitat spillover hypothesis'[91]: species flow between land-use types depending on their temporal and spatial requirements. This could then enhance local species pools and therefore increase β-diversity between forest locations.

Land-use not only shapes taxonomic communities, but also acts as an environmental filter for specific traits[55]. Large-bodied arthropods require larger patches of suitable habitat[59,99], and low disturbance regimes[60,61,100]. This is reflected in our results, where forests harboured communities with the largest arthropods, while the smallest ones were found in the most modified land-use types, in settlements. The most mobile arthropods occurred in arable lands, where species must adapt to the frequent cropping and replanting cycles with high mobility to escape disturbance and recolonise after[36,101,102].

Distance-decay patterns were mostly not impacted by body size. This implies that patterns observed at the taxonomic level are independent of the body size of specific species and preserved when body size as a trait determines community composition. Body size can be linked to many life strategies and behaviours adapted to local habitats[103,104], and depending on local species pools, some species could become more widespread than others[14,105]. Therefore, when local land-use types present complex habitat mosaics, highly frequent arthropod communities could become more different between each location. Studies on urbanisation gradients showed that insect communities are homogenised and shift to larger, more mobile communities[106,107]; however, we found settlements to harbour the most spatially heterogeneous communities of large species. This could be due to their large habitat-patch requirements and sensitivity to dynamically changing microenvironmental changes[59,60]. As a result,

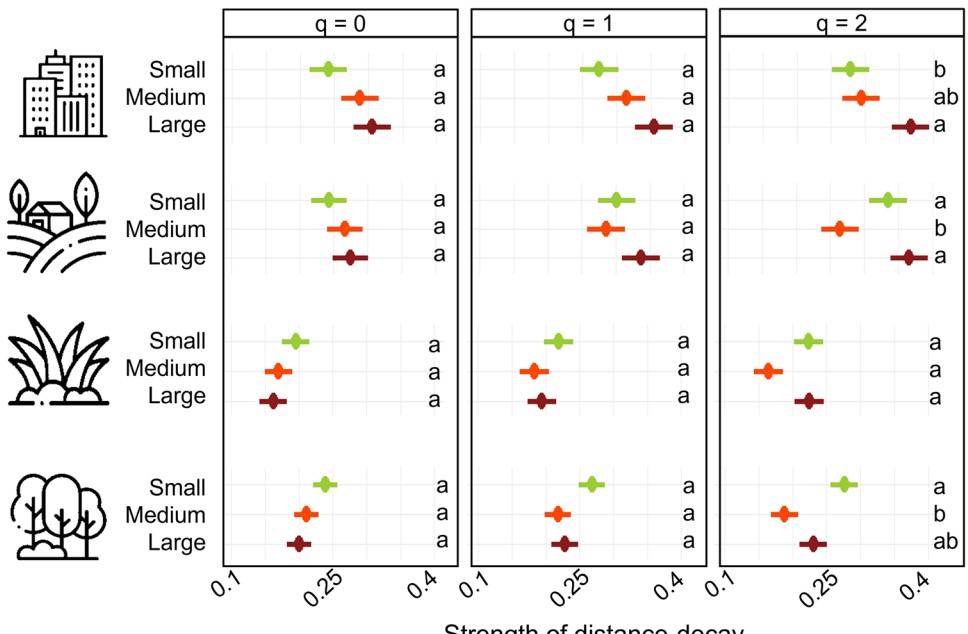

**Fig. 6 | The figure shows distance-decay of arthropod communities with differing body size categories.** The strength of distance-decay is represented by the absolute value of mean distance-decay curve steepness ± SE multiplied by 1000 in the four local land-use types: forest, $n = 55$ (forest green), grassland, $n = 46$ (purple), arable land, $n = 43$ (golden) and settlement, $n = 35$ (grey). The panels show insect communities focusing on species of different body sizes (lime = small, red = medium, brick = large) in the four local land-use types. Panels from left to right show communities over the Hill numbers: infrequent species: $q = 0$; frequent species: $q = 1$; and highly frequent species: $q = 2$. Significant differences are shown with different letters according to the SIMBA analysis. Icons designed by Mayor Icons and Freepik on Freepik.

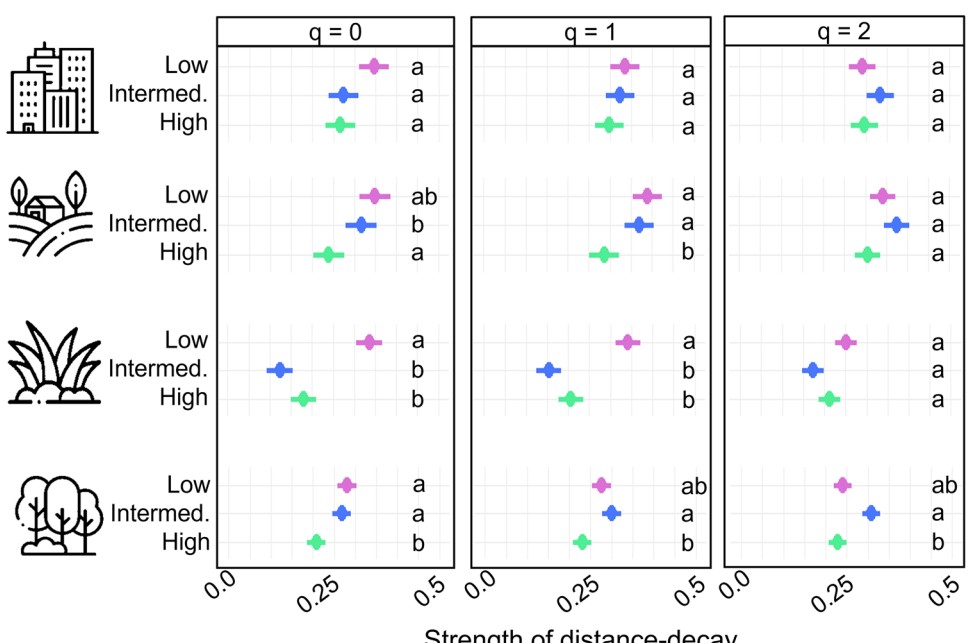

**Fig. 7 | The figure shows distance-decay of arthropod communities with differing mobility categories.** The strength of distance-decay is represented by the absolute value of mean distance-decay curve steepness ± SE multiplied by 1000 in the four local land-use types: forest, $n = 55$ (forest green), grassland, $n = 46$ (purple), arable land, $n = 43$ (golden) and settlement, $n = 35$ (grey). The panels show insect communities focusing on species of different mobility (navy blue = low mobility, violet = intermediate mobility, turquoise = high mobility) in the four local land-use types. Panels from left to right show communities over the Hill numbers: infrequent species: $q = 0$; frequent species: $q = 1$; and highly frequent species: $q = 2$. Significant differences are shown with different letters according to the SIMBA analysis. Icons designed by Mayor Icons and Freepik on Freepik.

highly frequent, large-bodied species can only occur in the less-modified pockets of a settlement, which are not evenly represented across all settlements. Forests, on the other hand, always exhibit low disturbance regimes and high structural diversity; therefore, arthropods might track microenvironmental differences, rather than distance itself [33]. In contrast, community assembly in forests might be more stochastic for small species, making these communities less similar between locations[108]. Body size only affected distance-decay when focusing on highly frequent species, which could be a spatial extrapolation to the findings of van Klink et al., where dominant species had the strongest responses to environmental changes over time[23]. In contrast to body size, distance-decay patterns of communities based on species mobility were affected by local land-use and species rarity. Forests showed a pattern where distance-decay depended both on species mobility and frequency. Although the patterns were not uniform, the disadvantage of low-mobility species in frequently disturbed land-use types[57,109] was clearly reflected in our results. Communities including low-mobility species were the most heterogeneous in arable lands and grasslands. Highly mobile species can escape the disturbance and rapidly recolonise after, while low-mobility species must find local refugia; therefore, populations might become very patchy, increasing heterogeneity between locations and strengthening distance-decay. Mobility did not affect distance-decay patterns in settlements. Mobility could be a poor explanatory variable in settlements, where species might select for the optimal habitat close to their origin, even if they are highly mobile[108]. Leaving an optimal habitat where greatly different habitats are very close together (a lush garden next to a parking lot) might be risky even for highly mobile species. Alternatively, species might be passively transported by human activities, which was shown among harvestmen (Opiliones) in an urban setting[110]. This process could result in mobility-independent patterns of distance-decay, and low-, intermediate- and high-mobility species could be equally likely to occur anywhere in settlements, ignoring the physical distance arthropods are able to travel.

As our major finding is the high homogenisation of arthropod communities in managed grasslands of Central Europe, it highlights the need for improvement in Europe's conservation actions[111]. While broad policies are needed for general directions, adaptive local management decisions impact diversity more profoundly[69,112]. Our studied grasslands and forests probably reflect the overarching problem of the long-term and mostly uniform forest and grassland management in Europe, which led to homogenous species communities. It is important to note that the current study did not include truly natural habitat types, and all studied plots were managed. There was no unmanaged natural baseline included in our study, which also reflects the lack of European natural habitats[113,114]. This urges the authorities to focus not only on the improvement of local habitat quality, but also on habitat heterogeneity between locations to boost regional diversity. Intense management practices in grasslands should be reduced by favouring extensive management practices (reduction of fertilisation, less intensive mowing), improving seed banks, or optimising grazing and cutting regimes[115-117].

Our study directly compared arthropod communities using distance-decay as a response to four terrestrial land-use types in Europe within the same ecoregion. The strong distance-decay in settlements and arable lands showed that these land-use types have a higher-than-expected heterogeneity between locations, contributing to overall γ-diversity. Thus, we found no strong evidence for broad-scale biotic homogenisation driven by anthropogenetic land-use, except for managed grasslands. The finding highlights the fact that anthropogenic habitat types can still create heterogeneous communities. Natural habitat types have been mostly lost in the Anthropocene, causing ongoing biodiversity decline, but human-modified habitat mosaics can have a beneficial impact on community heterogeneity. On the other hand, at a global scale, this effect disappears,

where net species diversity is declining. Management intensification on grasslands continues, which could lower the diversity of communities even more, while degrading multiple ecosystem services in the future. Permanent grasslands cover 13% of the area of the European Union (Eurostat, www.ec.europa.eu/eurostat) and still resemble a semi-natural habitat for many native species if managed extensively. Therefore, it is important to call for a change of management to halt further grassland community homogenisation.

## Methods
### Study design
The study was conducted in Central Europe in 2019 as part of the LandKlif project: https://www.landklif.biozentrum.uni-wuerzburg.de. 60 regions (each 5.8 × 5.8 km) were selected in Bavaria with land-use characterisations according to Corine land cover data[45]. Each region was classified into three regional land-use types based on the Corine land cover assessments: near-natural, agricultural and urban (to be called 'regional landscape). In each regional landscape, local land-use types were further determined based on the most dominant land-use type and vegetation using an area of 0.5 ha radius. Four local land-uses were determined: forest ($n = 55$), grassland ($n = 45$), arable land ($n = 44$) and settlement ($n = 35$). In each regional landscape type, three study plots were set up, in a combination of three different local land-use types, out of the four possible land-use types. This resulted in 179 study plots, covering a ~1000 m elevational gradient, mean annual temperatures ranging from 5 to 10.3 °C, and precipitation from 550 to 1961 mm, distributed in the whole state of Bavaria, Germany[45] (Fig. 2).

All plots were set up on a 3 × 30 m herbaceous vegetation near the studied land-use type. This was done to standardise traps in the four habitat types; while it possibly introduced some heterogeneity by establishing a small grassy area within forests, arable lands and settlements. However, we do not expect this to affect our distance-decay study, as the local land-use type should always dominate the habitats and drive arthropod species occurrence. Forest plots were mostly in managed forests, and broadleaf forest was preferred over coniferous ones. Study plots were at least 50 m from the edge, and in an opened, sunny position (e.g. forest glade). Managed grassland plots were established as part of larger managed permanent grasslands, as far as possible from other land-use types, at least 50 m. Arable land plots were set up adjacent to various field crops. Settlement plots were established in green areas with at least 50 m away from public roads and did not include forest patches.

### Sample collection
Malaise traps were used to capture invertebrates with the following dimensions: height front: 0.90 m; height rear: 0.60 m; length: 1.60 m, with 80% ethanol as preserving solution. Malaise traps are tent-like structures which capture arthropods by intersecting their ground movements or flights. Sampling containers are on top of the tent-like structure, using arthropods' tendency to move upwards[118]. Samples were collected, and a limited number of samples were already processed by Uhler et al.[65] using a slightly different approach in bioinformatics. However, for the current study, a new dataset was generated, including the full dataset, processed with uniform bioinformatic techniques. As with any sampling method, using only Malaise traps for arthropod sampling introduces a bias in sampled taxa and not capturing all arthropod taxa equally[119]. Nevertheless, this method reliably represents most groups of terrestrial arthropods[120], including flying and flightless groups (also shown in Supplementary Fig. 1): ground-foraging arthropods climb up on the net, and some small insects can be carried by the wind where the net intercepts this passive flight. Even though these traps are less effective in catching large flying insects with improved vision (such as dragonflies), they are still the most efficient trapping method for mass collection of arthropods[121,122]. Traps were active between April or May and August 2019, with a

fortnightly collection for 4 months, resulting in 8 sampling periods in total. The timing of sample collections varied due to weather and snow cover differences between locations. Samples were then split into two fractions, containing small and large species (or body parts) by sieving them through an 8 mm sieve to control for differences in biomass and to increase the identification of rare and small species[123,124].

## Next-generation sequencing

Following the fractionating step, the preservative ethanol was removed, and the mixed arthropod samples were dried overnight in a 60–70 °C oven to eliminate residual ethanol. The dried arthropods were then homogenised using stainless steel beads in a FastPrep 96 system (MP Biomedicals). DNA extraction from all samples was conducted by incubating them in a 90:10 solution of animal lysis buffer (buffer ATL, Qiagen DNEasy tissue kit, Qiagen, Hilden, Germany) with 10% proteinase K following the standard Qiagen DNeasy protocol. Following an overnight incubation at 56 °C, samples were cooled to room temperature. DNA was subsequently extracted from 200 μL aliquots using the DNEasy blood & tissue kit (Qiagen) according to the manufacturer's instructions. Amplicon PCRs employed 5 μL of extracted genomic DNA, Plant MyTAQ (Bioline, Luckenwalde, Germany), and HTS-adapted mini-barcode primers targeting the mitochondrial Cytochrome-Oxidase subunit I (CO1-5P) region (mlCOIntF/dgHCO2198; see Leray et al.[125]; Morinière et al.[124]; Hausmann et al.[126]; Uhler et al.[65]). Amplification success and fragment length (313 bp) were confirmed via gel electrophoresis. Cleaned amplicons were resuspended in 50 μL molecular water. Illumina Nextera XT indices (Illumina Inc., San Diego, USA) were added in a secondary PCR using the same annealing temperature but limited to seven cycles. Ligation success was verified through gel electrophoresis, and DNA concentration was measured with a Qubit fluorometer (Life Technologies, Carlsbad, USA). Samples were then pooled into 40 μL equimolar pools at 100 ng each, purified using MagSi-NGSprep Plus beads (Steinbrenner Laborsysteme GmbH, Wiesenbach, Germany), and eluted in a final volume of 20 μL. HTS was performed on multiple Illumina MiSeq (v3 chemistry, 2 × 300 bp, 600 cycles) runs with a target of 100,000 paired-end reads per sample. Six controls were included on each 96-well plate: 2 DNA-extraction controls, 2 PCR controls and 2 ligation controls to account for contamination or false positive results. Our initial target sequencing depth was 100,000 reads per sample. To address uneven sample coverage, we applied a normalisation approach that ensured each sample was analysed in proportion to its retained sequencing output, avoiding artificial rarefaction. This method preserves as much biological information as possible while maintaining comparability across samples. Importantly, all samples were processed at full sequencing depth rather than being subsampled to a uniform coverage threshold.

## Bioinformatics and taxonomic classification

Paired-end reads were merged using the USEARCH suite's -fastq_mergepairs utility in USEARCH v11.0.667_i86linux32[127] with specific parameters (-fastq_maxdiffs 99, -fastq_pctid 75, -fastq_trunctail 0). Adaptor sequences were trimmed using CUTADAPT v2.6[128]. Untrimmed sequences were filtered with the --discard-untrimmed option. Further pre-processing steps, including quality filtering, dereplication, chimaera filtering and clustering, were performed using VSEARCH v2.9.16[129]. Initial quality filtering was implemented with --fastq_filter (parameters: --fastq_maxee 1, --minlen 300). Filtering to a minimum length was applied, given the consistent amplicon length (313 bp) and subsequent quality control steps. After quality filtering, approximately 40,000 reads per sample remained, with ~30,000 reads retained after dereplication for downstream analysis. Dereplication used --derep_fulllength (parameters: --sizeout, --relabel Uniq), conducted at the sample level and then combined, with singletons excluded as noise. Following a 98% identity pre-clustering, chimaera filtering was done using the centroids algorithm and VSEARCH's --uchime_denovo utility.

This step reduced the size of the dataset and, therefore, the computational load by focusing chimaera detection on cluster centroids. While this method may slightly reduce sensitivity to rare chimaeras, subsequent quality control steps, including abundance filtering and taxonomic validation, help mitigate potential artefacts.

To minimise the impact of mitochondrial pseudogenes (NUMTs), we implemented multiple filtering steps, including de novo chimaera detection and removal, length-based filtering, and taxonomic validation using BLAST against a custom BOLD database and GenBank, in combination with classification via the RDP classifier. Taxonomic assignments were further refined through a consensus approach to minimise misclassification. Additionally, we implemented a stringent negative control-based filtering step: OTUs detected in any biological sample with read counts below the maximum observed in negative controls were excluded from further analysis. This step helps control for NUMTs that may arise from cross-contamination or environmental DNA artefacts. Given these multiple layers of quality control, we are confident that NUMTs have not significantly inflated the recovered taxonomic diversity in our study. Non-chimeric sequences were clustered into OTUs at 97% identity, with an OTU table generated through mapping reads to OTUs. False positives were minimised by excluding OTUs with read counts below 0.01% of total reads and by verifying OTU reliability against GenBank and BOLD databases via BLAST, following criteria for similarity and alignment quality. BLAST searches were conducted locally using the blastn utility from NCBI BLAST+ (ftp://ftp.ncbi.nlm.nih.gov/blast/executables/blast+/), with the following parameters: -task megablast -max_target_seqs 1 -max_hsps 1 -evalue 10 -word_size 28 -outfmt '6 sacc pident salltitles qseqid length'. For each OTU query, only the top match by percentage identity was retained. Searches were run against two separate BLAST databases: (1) the NCBI nucleotide (nt) database, downloaded from ftp://ftp.ncbi.nlm.nih.gov/blast/db/ (download date recorded in the result tables); and (2) a custom-formatted BLAST database constructed from COI records retrieved via the BOLD public API, parsed from TSV exports. Each BOLD record was annotated with a BIN (where available); BIN assignment was based on the best-hit from this custom BOLD database. This allowed us to maintain full control over query parameters and annotation consistency. The 0.01% abundance threshold was applied per sample, prior to taxonomic assignment. This conservative filtering step was used to minimise noise from index hopping, tag-jumping and stochastic PCR or sequencing errors, which could introduce spurious low-frequency reads[130–132]. Per-sample filtering ensures that true low-abundance OTUs in complex samples are retained if consistently present, while sample-specific artefacts are excluded. This approach is particularly relevant for large, heterogeneous datasets where global thresholds risk discarding real but rare taxa**[133]. Taxonomic classification employed RDP's Bayesian classifier[134] trained on the COI reference dataset[135] to complement our BLAST-based taxonomic assignments. The RDP classifier provides bootstrap support values for each taxonomic rank, offering a probabilistic measure of assignment confidence. By integrating RDP classifier outputs with BLAST results from GenBank and BOLD, we employed a least common ancestor consensus approach. This method ensures conservative and robust taxonomic assignments, especially in cases involving degraded sequences or incomplete reference databases. The resulting taxonomic summaries were visualised through KronaTools v1.311.

Taxonomic groups were then further resolved using the Barcode Index Number (BIN) system to define genetic units[136], with identification supported by the BOLD platform (www.boldsystems.org)[137]. BIN assignments were derived from BLAST results against a custom BOLD reference database built locally. COI sequences were downloaded via the BOLD public API in TSV format and used to construct a local BLAST database, in which most records included BIN designations in the fasta headers. After performing blastn searches, OTUs were assigned to BINs based on the top-hit match from this annotated BOLD database. Single

reads per BIN were removed, assuming sequencing errors. Due to gaps in existing libraries, assigning sequences to BIN units is challenging, especially for groups like dipterans, hymenopterans and hemipterans that often include dark taxa. We use the term dark taxa to refer to Barcode Index Numbers (BINs) or molecular OTUs that lack formal Linnaean taxonomic names due to gaps in reference libraries. This usage follows established literature where dark taxa denote unclassified or provisionally assigned taxa in molecular datasets[138,139]. These are common in metabarcoding of underrepresented groups such as Diptera and Hymenoptera, where many BINs remain unlinked to described species. For ecological analysis, we assigned sequences to units aiming to best represent species-level solutions and extracted ecological properties from sequence data. BIN alignment followed the protocol outlined in Müller et al.[22]. OTU sequences were aligned to the closest BINs in our custom BOLD reference database, which was locally constructed from BOLD COI data retrieved via the public API. For each OTU, we identified the top blastn hit with a BIN assignment and calculated the pairwise genetic distance. Sequences showing <3% uncorrected p-distance to a reference BIN were assigned as part of that BIN species; those >3% were considered genetic morpho-species. All alignment and distance calculations were performed locally using standard BLAST and sequence comparison tools, enabling consistency across the dataset and alignment to regional BINs relevant to our sampling scope.

## Statistical analysis

We used detection frequency (or incidence) in our dataset. Our data consisted of samples from 179 sites, with 8 sequential temporal repetitions. Then, each sample was divided into two fractions, depending on body size. This means that each BIN species had a frequency value (the count they occurred) ranging from 0 to 16 for all our 179 study sites. We used detection frequency instead of raw read abundance data for a stable metric for ecological analysis. Read abundances can be influenced by multiple biases, including sequencing errors, PCR amplification variability and primer-template mismatches[140,141]. This method should also control for the fact that arthropod populations can be highly patchy in distribution both temporally and spatially; therefore, abundance data could lead to species being over- or underestimated in each sample community. It has been shown that replicated incidence data support statistical approaches to diversity estimation that are just as powerful as corresponding abundance-based approaches[142,143]. Incidence data can be used instead of raw abundance data when the true number of individuals is not available[142], such as data derived from metabarcoding[133,144,145]. To handle the nature of incidence data, we use models specifically formulated for incidence data. In this model, only species detection/non-detection in any sampling unit or category is required; species abundances are not needed. While our frequency approach cannot estimate species aggregations[146], it does detect spatial and temporal community aspects better than abundance-based techniques[142]. Following the example of Colwell et al.[142], considering two species with the same number of individuals, but one species has a patchy distribution, and the other species occurs randomly. In this case, abundance-based techniques will be blind to association (and disassociation) and species aggregation. This topic is especially important in arthropod conservation. Optimal habitats are often patchy (for example, veteran trees) and can have a high number of specific species' individuals aggregated at a single tree. Even if there is a high local abundance of this species, it should be considered infrequent (or even rare), because it won't occur at any other habitat type or at another tree. In this case, it is important to use incidence-based techniques. In the current study, we distinguish between infrequent, frequent and highly frequent BIN species within each community of the four land-use types (see below in section Frequency). Nevertheless, it is important to note that frequency data represent the presence or absence of species. Frequency data ranged between 0 and 16: repeated temporal sample collection ($n = 8$) multiplied by two sample fractions based on body size (big and small, described in sample collection). In our incidence-based model, any species could be classified into one of the 16 possible categories of two size classes (big and small) multiplied by 8 sampling points in time. Body parts of big species (such as antenna) can be sieved into the small fraction, and small species or body parts could end up in the 'big' fraction attached to big species (e.g. mites).

All statistical analysis was done using R statistical software, version 4.3.2[147]. First, we assessed sample coverage of each community, then standardised the samples, see (i) Sample coverage. Then, we had three components of the statistical analysis: (ii) Frequency: establishment of community similarity matrices along the Hill numbers[38], where species are weighed differently based on their detection frequency, giving them a 'rarity' variable (see below). Then, (iii) Traits: creation of trait-based community matrices, where species are categorised into trait groups according to their body size (large, medium and small) and dispersal ability (high, intermediate and low-mobility) using the existing literature and expert knowledge. Lastly, (iv) analysis of distance-decay patterns: model distance-decay relationships in the four land-use types according to their rarity and traits (Fig. 1). All graphs were created with the ggplot2 package[148].

**Sample coverage.** Measures of β-diversity (and distance-decay) are calculated by gamma-diversity divided by alpha-diversity, which means that this metric depends on between-sample differences and therefore is dependent on sample coverage. However, it is shown that most samples are incomplete, thus standardising sample coverage becomes very important[42,149]. 'Sample coverage' was originally developed by Alan Turing in his cryptographic work, which is defined as the proportion of the total number of individuals in the entire assemblage that belong to the detected species in a sample. Chao et al.[150] extended Turing's concept of sample coverage to its incidence version to quantify sample completeness of replicated incidence data. This extension is based on the distribution of incidence frequencies obtained from multiple sampling units (such as DNA samples with identified BIN species). In our study, sample coverage may differ between local land-use types; therefore, we used coverage-based indices calculated by the package iNEXT.beta3D[40], which enables us to compare samples from different land-use types without any detection bias in our samples.

**Frequency.** Based on the framework of the effective number of species, the [38,151], we can further consider the rarity of BIN species within communities. We can shift the focus of diversity from infrequent species to highly frequent species by adjusting the diversity order q when creating similarity matrices by taking relative frequency distributions into account. Originally, Hill numbers were developed for abundance data; this approach was extended to replicated incidence data[142,151]. Under the adapted model for incidence data, only species incidence (detection/non-detection) in any sampling unit is required; species abundances are not needed. As with abundance data, Hill numbers based on incidence data can be similarly formulated by replacing species relative frequencies/abundances with species relative incidence frequencies. For $q = 0$, the incidence-based Hill number reduces to species richness and thus is more sensitive to infrequent species; the measures with $q = 1$ and $q = 2$ can be interpreted as the effective number of frequent and highly frequent species in the community, respectively. Species matrices based on their frequency were created using their overall frequency in the dataset. This variable is purely based on detection frequency within a community, and did not take their functional contribution or geographical range into account[152]. This way, all frequency classifications are sample-dependent, as infrequent, frequent and highly frequent categories were assigned to species compared to others within the same sample.

Therefore, the same species could be infrequent in one sample, but highly frequent in another one. All categories refer to the sample-scale frequency, i.e. local infrequent, local frequent and local highly frequent species within the sampled community. We constructed community matrices along Hill numbers using standardised coverage-based indices for each land-use type, body size and mobility. Matrices were created using data type = 'incidence raw' within the iNEXTbeta3D function. Using the above-described similarity distances, we created community similarity matrixes for rarity ($q = 0$, $q = 1$, $q = 2$) within the four land-use types, resulting in 12 subset matrices.

**Traits.** We used body size and mobility of 450 arthropod families from classes Arachnida, Chilopoda, Collembola, Diplopoda, Diplura, Insecta and Malacostraca (Isopoda). Traits were assigned at the family level (averaged where possible) using existing databases, expert opinions and taxonomic keys (sources can be found in Supplementary Data 1). Body size classes were established based on the distribution of the data, instead of a predeterministic approach. The classes were determined based on the body size minimum and maximum values of the occurring BIN species. We considered arthropods as small if body size was equal to or less than 2.7 mm ($n = 3517$), intermediate between 2.8 and 7 mm ($n = 3493$), and large equal to or above 7 mm ($n = 3578$). 91% of BIN species were classified for body size. We used mobility as a category, which was determined by experts: low ($n = 1255$), medium ($n = 2516$) and high ($n = 6840$) (for detailed classifications, see Supplementary Data 1). To confirm the body size categories, a subset of measured body sizes of species belonging to 71 families[153] were tested against our estimated body size categories, and revealed a close match between these families, supporting the robustness of our approach (Supplementary Fig. 4). Average body sizes and mobility scores were calculated using the package FD[154] as community weighted means for each land-use type. Pairwise differences were tested with an analysis of variance and Tukey's post-hoc test using packages lme4[155] and multcomp[156].

**Analysis.** Distance-decay is defined by species community similarities (or dissimilarities) between two given study sites (or samples) and the physical distance between these two sites, pairing all studied samples and locations. This way, we created two matrices for every rarity and trait group: a community similarity (ranging from 0 to 1) and a physical distance matrix (ranging from 0.3 to 482.4 km), where the x- and the y-axes are the paired sampling sites, and the cells are the community and physical distances between them.

Community similarity matrices were created according to rarity (detailed above) and traits. BIN species belonging to trait-classified families were assigned to a trait group. Then, different matrices were created, including only specific trait groups according to body size and mobility (3 matrices each). All trait groups were then reassigned to the land-use types they occurred in, resulting in trait group subsets for each land-use type (3 body size matrices and 3 mobility matrices for each land-use, 24 subset matrices in total). All groups had their corresponding rarity submatrices, $q = 0$, $q = 1$ and $q = 2$.

Slopes of distance-decay relationships were compared between land-use types, mobility and body size classes. Given that community similarity matrices were derived from sample coverage-based diversity metrics according to Hill numbers, distance-decay relationships were compared with the package Simba[157], which accepts a variety of similarity matrices instead of creating one based on fixed indices (such as Sorensen or Jaccard). For significance tests, we used the function 'diffslope', which compares slopes of different distance-decay regression lines directly. The function uses a randomisation approach, comparing our datasets instead of a bootstrapping approach based on a simulated dataset by model parameters. We used the Bonferroni–Hochberg correction for multiple comparisons between the four habitat types. Furthermore, we constructed linear regression models and extracted the distance-decay slopes for better visualisation. Main outputs from linear regressions can be found in the Supplementary Tables 1–4.

To test whether regional landscape types change the general patterns found within local land-use types, we compared slopes of overall land-use types and specific land-use types surrounded by each regional landscape (urban, agricultural or near-natural). This was done by deducting the absolute value of the general slope of local land-use (including all landscapes) from the absolute value of the slope of the same local land-use but imbedded in a specific landscape type. This way, positive values imply an increased heterogeneity (more negative distance-decay slope) of a given regional landscape type compared to the local slope, which did not consider the surrounding regional landscape type.

### Reporting summary

Further information on research design is available in the Nature Portfolio Reporting Summary linked to this article.

## Data availability

The datasets on arthropods and traits generated in this study have been deposited in the Figshare database[158]. The trait data generated in this study are provided in file Supplementary Data 1. DNA sequence data of OTUs used in this study are available in the NCBI's SRA database [www.ncbi.nlm.nih.gov/sra] under the accession code PRJNA1345334, Biosample: SAMN52666869. Data are provided with this paper and available at the above link in the folder 'used_datasets.zip'.

## Code availability

The used R code is available via the Figshare repository[158] under the file name 'Distancedecay_Incidence_finalSubmission'.

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

## Acknowledgements

The authors would like to thank all landowners who contributed to this study. The authors acknowledge the support of all students and technical staff in the field and laboratory. The authors thank Torsten Hothorn and Oliver Mitesser for statistical support, and Caryl Benjamin, Rebekka Riebl, Sandra Rojas-Botero and Lars Uphus for fieldwork and technical support. This study was conducted within the framework of the joint project LandKlif (https://www.landklif.biozentrum.uni-wuerzburg.de/) funded by the Bavarian Ministry of Science and Arts via the Bavarian Climate Research Network (bayklif). Open Access funding enabled and organised by Projekt DEAL. Icons used in illustrations were made by Freepik (http://www.freepik.com/) authors: Fig. 1. arthropods: diplopod: afif fudin, wasp: Freepik, grasshopper: Nadiinko, ladybug: iconfield, beetle: Nikita Golubev; land-use types Figs. 1–7: forest: Freepik, grassland: Mayor Icons, arable field: Freepik, cityscape: Freepik.

## Author contributions

O.D.: conceptualisation (lead), analysis (lead); writing—original draft (lead); writing—review and editing (lead). J.U.: investigation (lead). S.R.: design (lead), conceptualisation (equal); investigation (equal); project administration (equal); review (equal). A.C.: analysis (equal), review (equal). M.K.: analysis (equal). I.D.: project administration (equal); review (equal). C.T.: review (equal). J.E.W.: review (equal). J.E.: review (equal). U.F.: review (equal). C.G.V.: review (equal). M.H.: review (equal). V.B.: analysis (equal), review (equal). J.M.: analysis (equal), review (equal). J.Z.: conceptualisation (equal). J.Mü.: conceptualisation (equal); formal analysis (equal); methodology (equal); project administration (equal); supervision (equal); review and editing (equal).

## Funding

## Competing interests

The authors declare no competing interests.
