## [Transparent Peer Review file · Nature Communications]

Distance decay reveals contrasting effects of land-use types on arthropod community homogenisation

Corresponding Author: Dr Orsi Decker

Version 0:

Reviewer comments:

Reviewer #1

(Remarks to the Author)

Decker et al. explore the effect of land use on biotic homogenization in arthropod communities using a large metabarcoding dataset from Malaise traps in Bavaria. They sampled arthropod communities in four land cover types of differing land use intensity, forest, arable land, human settlements and grassland and explored the community's distance decay. When incorporating information on species abundances, as well as body size and mobility, they find that grasslands are the most homogeneous habitat type. Contrary to expectation, settlements and arable lands showed the spatially most heterogeneous communities.

The work follows a very interesting and timely research question and I must commend the authors for the large amount of work put into this. While the manuscript is generally well written, the text contains various typos and grammatical mistakes. Unfortunately, I also see several issues with the study, which question the validity of the presented results. My two main qualms are:

1) Methodological details are lacking to judge the validity of the data: The study is based on a large DNA metabarcoding dataset and understanding this data is essential to judge the validity of the presented results. However, the methods lack all important details to judge on the reliability of the data. Considering that metabarcoding data is the core of this entire study, it is not sufficient to simply cite another study to provide the methodological approaches (the cited study also lacks many important details). The minimum information, which needs to be provided in the methods and results of the work are: What kind of field and lab controls were used and how were contaminations accounted for? What PCR and/or extraction replication strategy was used? What sequencing coverage was aimed for and were any libraries undersampled? The study cited as methodological reference refers to only a single Miseq V3 run. This would be by far not enough for a dataset as big as the one presented here. What was the taxonomic composition of the recovered samples, how comparable were the communities between habitat types? Why was no analysis performed for different taxonomic groups, for example orders? Considering the authors used Malaise traps, I assume there will be very dominant Diptera and Hymenoptera representation in the data.

2) The ecological annotation, which is core to this work, is not reliable: The results are based on a classification of the recovered OTUs into abundance, body size and dispersal categories. The validity of these classifications is essential for this work. Abundance is identified based on repeatability of detection in 8 consecutive trapping events over 4 months. Using repeatability of occurrence is a great idea, but has not been properly tested in arthropods. But the main issue here is that the authors do not identify abundant species with their approach, but species with a temporally stable occurrence. Arthropod communities can show pronounced temporal community turnover in 4 months. Provided the sequencing was deep enough, the authors will detect even very rare species in each Malaise trap. Using the author's approach, rare species with stable occurrence over time will be wrongly assigned as abundant. At the same time, highly abundant species, with a short seasonal occurrence would be judged as rare. You would need replicate traps at the same time point to apply the approach used by the authors here. I also feel that assigning body size for entire families is an oversimplification.

Please find further comments below:

The authors refer to species throughout the text, for example, the abstract refers to 12,000 species. I feel this is misleading. They use 3 % radius OTUs for their work. OTUs can overestimate the true species diversity by severalfold, this will also

happen with BOLD BINs. It would be better to refer to OTUs in the text.

The metabarcoding methodology is very critical to judge the validity of the work. I do not feel it is sufficient to cite Uhler et al. here. Uhler et al says their data was sequenced on one MiSeq v3 kit. This would be way too little data for a dataset of so many Malaise traps, as you present here. Recent work suggest sampling to several 100,000 to even millions of reads for a proper recovery of taxa in a Malaise trap.

I am not sure is this study is based on data from a previous study (Uhler et al.) or if the data was generated for this work exclusively? If so, mention clearly in the methods

L362: "Nevertheless, this method reliably represents most groups of terrestrial arthropods, including flying and ground-dwelling group"

I beg to differ here. Malaise traps are very biased towards Diptera and Hymenoptera. For example, most Malaise traps are unsuited to sample beetles. Malaise traps are so popular, because 1) insect decline was shown with them and 2) because they sample hyperdiverse Hymenoptera and Diptera groups, hence a very large species diversity. But not necessarily an exhaustive and representative spectrum of a community.

L369 what is COI 5P? I assume it refers to the 5-prime region of the COI gene. You could just say the COI barcode region.

L380 – 381 I assume the first one is supposed to be < 3 %

L386 repeating the same info twice

L392: change to "abundance data"

L395: "By using frequency, we consider species captured at all sampling periods (captured 16 out of 16 times) as species with high relative abundance"

As stated above, this is not true. See my explanation above. You would need local subsampling, e.g. replicate traps from the same time window. As your entire study is based on this assignment, this is a big problem.

L430 traits: body size was assigned at family level. I feel this an oversimplification. This may work in some abundant families like Cecidomyiidae or in whole groups like Collembola or Acari. But many arthropod families show massive variation in bodysize. This is also true for the hyperdiverse Hymenoptera and Diptera dark taxa which dominate Malaise traps.

Traits: Your dataset is probably very biased towards Hymenoptera and Diptera. How do the observed patterns change between different taxonomic groups? Why do such a taxonomically coarse analysis, if barcoding allows identification of all groups to fine scale taxonomic levels?

Also, was your trait datasets taxonomically biased? E.g. were certain taxa/orders overrepresented in the small or big or high or low dispersal category? Could taxonomy be a driver of the observed effects? How did you control for this?

L432: why do you have hexanauplia in your data? Aren't they only aquatic arthropods?

L478: Data availability: Metabarcoding data need to be available, that includes OTU tables as well as all raw reads.

Reviewer #2

(Remarks to the Author)

Review Nat-COMM-24-352777-T

Review summary

The paper is overall well-executed, with a clear and compelling presentation of the research context. While the topic itself may not be entirely novel, the study distinguishes itself by applying innovative fieldwork techniques to test new hypotheses within the field. This adds significant value and relevance to the existing body of knowledge. The authors have effectively articulated the significance of their work, making it a valuable contribution to the ongoing research in this area.

In their study, the authors investigate the impact of landscape homogenization across distances on insect diversity loss by examining four distinct landscape types: forests, managed grasslands, arable lands, and settlements, all within near-natural, agricultural, and urban areas. They hypothesized that the decline in beta-diversity over space (distance-decay) would intensify with increased land-use intensity. Focusing on arthropod communities, however, they found results contrary to their expectations. The most homogenized communities were observed in grasslands, while the greatest diversity between locations was found in settlements and arable lands. Beyond community homogenization, they also explored species traits within these communities. As anticipated, larger, and less mobile species exhibited more spatial heterogeneity than smaller, more mobile species. The title of the submitted article highlights the complexity of these findings. Additionally, the article delves into these results, which contradict expectations and previously published literature, offering insight into the complexities of land use management and its effects on arthropod communities.

While the manuscript has significant strengths and addresses a highly interesting topic, it requires major revisions to

enhance clarity and emphasize the main message. The finding that the most anthropogenic environments remain more heterogeneous than the 'most natural' ones is particularly unexpected, yet it is not sufficiently highlighted in the abstract or discussion. Given the complexity of the study, which spans various scales (both temporal and geographical), land-use types, species rarity, and traits, I recommend creating a schematic representation of the study to help readers follow the concepts more easily (an improved version of Figure 0). Additionally, I suggest revisiting some discussion points for better emphasis and possibly reorganizing the discussion to present the ideas in a more logical sequence. Lastly, several minor revisions are needed to correct sentence errors and improve overall readability. All comments and suggestions are provided in detail in the following review.

Major issues

First, as outlined in the summary, the main messages regarding the complex diversity response to different land-use types are not clearly emphasized in your abstract and discussion. As a result, the reader may struggle to grasp the key takeaways. This may be due to the inherently complex structure of your study, which is indeed a strength. However, improving certain sentences and refining Figure 0 could significantly enhance the clarity and coherence of the main text. I suggest revising Figure 0 to align with the various steps of your study as described in the Methods section (i, ii, iii). This will help clarify the scales of analysis, which can be somewhat confusing—particularly when distinguishing between the local level (four types of environments) and the regional scale (near-natural, agricultural, urban). Additionally, some of the minor, line-by-line comments may further assist in addressing these points.

Overall, the text could be made clearer by more precisely defining terms used throughout. For example, please clarify whether 'strong distance decay' refers to 'high heterogeneity,' and then consistently use one term to avoid confusion. The same applies to the abbreviations for beta and alpha diversity—define them clearly and stick to one throughout the text. Additionally, the phrase 'on a scale' is frequently used; however, 'at a scale' may be more appropriate in most contexts. For instance, it's correct to say, 'on a scale from local to global,' but 'at the global scale' is more accurate.

Please also ensure consistency in the use of either American or British English throughout the manuscript. Furthermore, some sentences are quite long, making it difficult for readers to grasp the main points quickly. Examples of this are provided in the line-by-line comments. Finally, I suggest making your Conclusions more concise. While all the necessary points are present, the lengthy sentences can obscure the main message, making the text and its justifications harder to follow.

Finally, a few additional points that could enrich the discussion are worth considering. These are detailed in the line-by-line comments, but for instance, there is limited discussion on forest composition, despite noting a potential bias in the methodology. Similarly, the grassy areas within settlements have been mentioned, but could benefit from further elaboration.

Minor issues and line-by-line comments

I.1. Prepared for Nature Ecology and Evolution?

I.20. Verify affiliation, Lepizig? Guess it is Leipzig

I.22. Please remove one repeated 'during'

I.30-31 Please rephrase, what is tested is unclear here

Figure 0: I suggest adding directly on the figure that left distance decay represent Hypothesized distance decay whereas on the right it is the actual result of the paper.

I.50. I disagree with the fact that specialized species decline in human-modified habitats. In fact, some very specialized species are selected. Some others are counter selected. Maybe you can be more specific here or modify your sentence. This is explicitly the title of the first article you cite.

I.51. More or less

I.52 and 54. Beta-diversity, please define and use the same expression everywhere

I.62. Maj is missing. Also stick to one term: homogenization or homogenisation?

I.74. Various different?

I.73-76. Here I do not get the main idea you are talking about. Are you saying that studies at global scale cannot conclude on the factors that contribute to communities' changes? Are you saying that studies looking at the effect of management of land-use specifically are scarce? Please rephrase.

I.78. Please add a link with previous paragraph.

I.80. What are Malaise traps? Maybe cite a research work on that.

I.81. What do you mean by macro-environmental differences? What were you looking at specifically?

I.82 on?

I.82-84. Unclear what is the local scale, what is the regional scale? Some precision might be needed for a better understanding.

I.82-86. Very unclear here, it is a bit confusing to talk about local scales/regional scales and then local land-use types. Can you please rephrase?

Figure 1b. Can you change the map font, for more visibility?

I.109. What are time periods? Until then it is nowhere mentioned?

I.110. What is BINs the acronym for?

I.114 to 116. This is an interesting result around species richness that can be discussed.

Figure 2. Why are the index value to calculate distance decay different when looking at different species type? Please help the reader (although explained in Methods).

I.154. Not significantly different from grasslands that also records small species on average.

I.155. Maybe this is interesting to note that arable land might be heterogeneous because they record the most mobile species among their communities?

I.157. Even if we may know that species size has something to do with species life history traits, dispersal, or mobility, I think there is a need to explain why small or large bodied species are expected to behave differently, as finally you do not find any

difference (l.157).

Additionally, I find the paragraph on body size and homogenization in different land use type a bit hard to read, is not there another way to say the things? Large species are more homogeneous in settlements, arable lands, and forest for typical species. Additionally, nothing is discussed here for grasslands.

Figure 5. No change in communities' composition for rare species whatever size and habitat. It could be nice to add something pondered by the richness in the different land-use type? Maybe it could be nice to help the reader understand the figure to add an axis how to read heterogeneity and homogeneity along the distance decay axes.

L.184. Less community similarity: maybe define appropriately homogeneity and heterogeneity once and then stick to that term. Sometimes you say more heterogeneity, sometimes less similarity. Maybe on that figure you can also highlight that grasslands are very often more homogeneous than the other land-use type. I recommend revising this figure to simplify its interpretation and clearly convey the key messages.

L.169. Something here interesting on the fact that intermediate mobility species seems more heterogeneous in community than low and high.

Figure 6. From this figure, I would primarily conclude that insects with low mobility differ between locations with grassland and forest land use types, which is quite interesting. It appears that the dispersal pattern is only evident in the more "natural" environments, such as grasslands and forests, and not in arable lands or settlements. This observation could be explored in relation to exotic species, species dispersal abilities, and species richness at these locations.

L.197. What is the difference between this lines and l.76? Same sentence, same papers. Additionally, this citation style is somewhat awkward.

L.199. see is not needed.

L.196 – 200. A lot of "but". Please rephrase this paragraph to be more direct and break it into separate sentences.

L.201-202. "Regional and local diversity assist in conservation planning"? "designs of available data often do not allow for testing » could you please rephrase in something more direct? Something like: Conservation planning is based on regional and local diversity. However, study designs that offer available data for testing diversity at the appropriate scale are often lacking.

L.204. Can you improve the phrasing?

L.207. No need for "see also"

L.215. Some discussion might be needed on that point. When looking at mobility, low mobile species are not more heterogeneous between locations in land use type arable and settlements whereas there is a difference in grasslands. So maybe something has happened in those areas regarding low mobile insects? What I mean here, is that with distance between location, you may expect low mobile species to differ more than highly mobile species.

L.228-229. You might also discuss how agricultural practices and pesticide use significantly contribute to the reduction of various insect species in certain arable regions, while these practices may differ substantially in other areas. This could explain or support your findings of greater heterogeneity in areas with more intensive land use.

L.232. What is reflected in your study? You are talking about arable field size. I guess you are mentioning the fact that the intensity of arable land use is less important than in Germany. Please rephrase and justify. The justification follows subsequently. I suggest something like: In accordance with recent studies, our results confirm the importance of high crop heterogeneity and field size on biodiversity.

L.235-239. Well, you never mention what is alpha diversity in your article. So please explain it somewhere or rephrase. It is hard to understand how you can compare alpha diversity (within location diversity) and your distance decay based on beta diversity (between location diversity). Please explain and rephrase this paragraph.

L.242. alpha-diversity of arthropods species richness, seems that you repeat or misuse the terms. Alpha diversity is already a measure of the species diversity OR richness.

L.249-252. Interesting part of the discussion, but it is not what you observe for grasslands except if I misunderstood?

L.254-255. passive sentence please rephrase.

L.258. Do you mean forests isolated by urban environments? If yes, please rephrase.

L.317. "to four"

L.320. Now is gamma diversity that has not been defined earlier ... I suggest you define beta, alpha and gamma in the introduction.

L.322-324. Long sentence a bit hard to follow. Please rephrase and please reword "the fact, that however"

L.325. A lot of "however" in the justification. Please consider rewording.

L.339. It is very clear in the methods. This is not consistently the case throughout the text, which might become clearer with revisions.

L.340-341. Please consider rewording this sentence.

L.341. on a resolution?

L.342. I assume you are still referring to regions. It is important to define one term and consistently use it throughout the text, as the current wording makes it difficult to discern the scale you are discussing—whether it is regions, plots, or land use types. This issue recurs throughout the article, and maintaining clarity would greatly benefit from consistently using three main terms.

L.346. "for details see"?

L.352. A variety of forest types are covered in your study, but this fact is not discussed anywhere in your article in regard with your result. Although you discuss a lot the variety of crops that may enhance heterogeneity in your arthropod's community, you do not discuss why arthropods community are not that heterogeneous in forest? I may have the same remark for settlement plots that are taken in green areas, that may largely vary from one settlement to one another in term of plant species, exposure to the sun, temperature ... I think something interesting is missing in the discussion of your research.

L.380-381. Please reword not using mathematical abbreviation. Also, you are using two times the same reference to superior to 3 percent (>3%) but say "while" so I guess one should be inferior to.

L.386-387. You say two times the same thing in two different sentences.

L.389-390. Reading this sentence, I do not understand which method has issues. "This method".

I.392. Can you please be clearer on the fact that abundant data may lead to underestimated species?
I.396-397. I do not understand here what are the fractions you are talking about?
I.398. In the method you start talking about sample and now you use assemblages. Maybe define that sample and assemblages are the same at the beginning and then only use sample?
I.402-404. Can you be clearer?
I.405. Space missing when starting new sentence.
I.419. Mistake in the bibliography citation?
L.423. Maybe "however" is not the appropriate word here to justify your idea. "Although" might be better?
I.413-429. On average, this is clear, but a formula might better illustrate what are those "q" value? In general, as your method is complex in term of duplication/steps, it might be a good idea to add a Figure (supplementary or in the main paper, as Figure 0 may be improved) so that everything is clear for the reader. It is hard to remember the scales, the index and so on.
I.436-437. What were your criterion to decide this classification. It seems this is only based on your data and not on physiological value or other biological explanation. Could you please specify just with a small sentence in addition to the supplementary?
I.465. Please remove the extra point in the middle of the sentence.
I.492-493. Maybe remove "on their land", it is a bit obvious.

Additional remarks

- Why do you have doi for some papers and not for others?
- General remark for figure, maybe change $q=0$ for rare species in the figure instead of explaining what it is in the caption.

Version 1:

Reviewer comments:

Reviewer #1

(Remarks to the Author)

I'd like to commend the authors for the significant revision of the manuscript. As stated in my previous revision, this is an interesting and timely manuscript. Its also nice to see metabarcoding being used for consequential ecological questions. The manuscript has improved, especially relating to the details of the molecular method. However, I still have several issues relating to the molecular methodology and especially relating to my earlier comment on using detection frequencies as a proxy for abundances or rarity.

1) While the presentation of the molecular data has considerably improved, I'm still missing details here. The authors have provided a sophisticated statistical analysis and explained it in great detail. But the analysis will only make sense, if the underlying data are reliable. Readers need to be able to judge the reliability of the data based on the manuscript. I still can not fully do this right now. I'll explain what's still missing:

A detailed treatment of the negative controls run along all samples is lacking.

L464-466 "False positives were ... removed below 0.01% of total reads and by verifying OTU reliability against GenBank and BOLD .."

This is all, which is explained and a good first step. But this is not enough to remove false positives. You should use the negative control to inform about the level of background contamination. You say you have six controls per plate, but then never talk about them anymore. How were the negative controls used to inform data cleaning? Please explain in more detail. How did you deal with NUMTs? Is it possible that NUMTs inflate the recovered taxon diversity for groups with a high NUMT frequency?

You aimed at 100 k reads per samples, but what was the achieved coverage per sample in the end? How did you deal with uneven sample coverage between samples? Where all samples sequenced to exhaustion, or did you rarefy the data to some coverage level? Please show/explain this in the manuscript.

2) The other issue remaining is the use of frequency information to distinguish rare from common taxa. You are citing references 129-131 to justify your approach. I have no doubt that frequency data is very useful in ecological analysis. I even think using frequency of occurrence is probably is a good way of approximating abundance classes from DNA data. But I'm not sure, if this has been properly tested yet. References 129-131 are all references from the general ecology literature. None relates to metabarcoding data. Please cite metabarcoding literature, which suggest using frequencies as an abundance/rarity substitute.

You argue that by dividing each sample in to two size classes you double your replication. I do not think this is true. Species which occur in both size classes are not frequent species, but species, which have a wide range of bodysize. This could for example be a species, which you find as adult and larva in the same trap. Species in only one size class are those which are only big or only small.

Also, the issue still remains that using frequencies of occurrence across 8 time intervals does not necessarily indicate frequent species, but species with a temporally stable occurrence. So, your way of using frequency of occurrence across 16 subsamples across several months per site will not show common vs. rare species. Instead, it will show species with a wide size variation (e.g. different ontogenetic stages) and long-term temporal occurrence. Some flies, for example, can be hugely

abundant at a site, but only for a few weeks. A species like this would be rated rare in your approach, because you only find it in one time window. Please explain how this is not affecting your inferences in the methods and discussion.

One way of finding out if the classification of rare and frequent species is correct, would be to explore the actual taxon lists. You could provide a table with species in the different rarity classes and explain, if the grouping makes sense. After all, the authors seem to know their ecosystems very well. So, they should be able to show with their data that the classifications for species into rare frequent is actually true.

L411-412: "Samples were collected, and a limited number of samples were processed already by Uhler et al., 2021. However, for the current study, the full collection was processed."

So, the data was generated newly for this work, but the specimens were collected by Uhler et al. 2021 already? Then state it explicitly.

Supplementary Figure.1. Please provide more detail in the caption. I can not follow what is exactly shown here, please explain better. 1 a) average frequency per plot? Is this the number of OTUs found per plot for each taxon? Why would Dermoptera be present in so many OTUs at the plots? Isn't there only a handful of species for the group in Germany?

L604 Data availability: A link is missing here to the OTU tables and raw sequence data.

(Remarks on code availability)

Reviewer #2

(Remarks to the Author)

Review Nat-COMM-24-35277A-Z

Reviewer Response Summary for Decker et al.

The study aims to investigate the role of land use in shaping beta diversity patterns of arthropods. The authors analyze arthropod communities based taxonomic diversity and on functional traits (body size and mobility) along a land-use intensity gradient within three different landscape types, employing distance decay analyses. The study stands out due to the extensive fieldwork conducted by the authors, who implemented large sampling techniques to collect a comprehensive dataset. Additionally, the research benefits from a robust laboratory pipeline, which allows for precise taxonomic identification and trait analysis. This combination of large-scale field sampling and cutting-edge genetic methodologies enhances the study's impact, providing valuable insights into the mechanisms driving community homogenization. Their hypothesis suggests that more natural habitats should exhibit greater community heterogeneity, while those embedded in highly human-modified landscapes should show increased homogenization. Focusing on arthropod communities, the study revealed unexpected patterns, the most homogenized communities were found in grasslands, whereas settlements and arable lands exhibited the highest diversity between locations. Beyond community homogenization, they also explored species traits within these communities. As anticipated, larger, and less mobile species exhibited more spatial heterogeneity than smaller, more mobile species. The title of the submitted article highlights the complexity of these findings.

The manuscript has benefited significantly from the reviewers' comments, and the authors have addressed these points effectively. They have provided additional justifications, analyses, and explanations that strengthen their study, adding depth to their findings. The aims of the study are now more explicitly stated in the introduction, improving clarity. The authors have now provided a more precise explanation of their expectations regarding species trait homogenization, particularly in relation to body size and mobility, with better contextualization within the existing literature. The discussion section is now more structured, making the arguments clearer and more cohesive. The inclusion of supplementary materials is particularly useful, offering valuable background on the study's vocabulary and methodology. The newly structured Figure 1 is greatly appreciated, as it aids in navigating the study's complex data and analytical framework.

My main remaining remark concerns the need to clarify and homogenize the terminology throughout the text. Although the authors have made a huge effort regarding the vocabulary and provided a terminology box, there remains some inconsistency in the use of key terms, particularly "distance decay," "species homogenization," and "community similarity loss." While "distance decay" refers to changes in species composition along a spatial gradient, "community similarity loss" seems to convey a similar concept. To enhance clarity and consistency, the authors should decide on a single term or explicitly define their usage of these terms within the text. In the results section, "community similarity loss" is used, while the discussion section references both "distance decay" and "species homogenization." The interchangeable use of these terms may lead to confusion. A clearer, more uniform application of terminology would help reinforce the study's conclusions.

Minor issues and line-by-line comments

l.20. Please correct to Leipzig.

l.56. When are you referring to your vocabulary box?

l.103 to l.105. Here you do not describe what are your "local land-use" and "regional landscapes". Maybe you could briefly name them here and relate them to your Figure 1?

l.155. The term community similarity loss feels ambiguous—could it refer to species turnover or simply a reduction in species richness? To me, this phrase seems new and unexpected in the results section, as it was not introduced or defined earlier. I

understand it relates to the concept of distance decay, where species composition changes along ecological gradients—communities farther apart in space share fewer species compared to those closer together.

Additionally, while I recognize that you utilize various beta diversity measures to analyze changes in species composition from less frequent to more frequent species, I am not fully convinced that community similarity loss is the best term to describe these patterns. Wouldn't it be simpler and more intuitive to frame this as heterogeneity and homogeneity?

Alternatively, you could describe it as decay in community composition. For instance, a steep slope would indicate high decay, representing more heterogeneous communities across distance, while a gradual slope would reflect slower decay and thus more homogeneous communities along the same distance gradient.

I suggest clearly defining your preferred terminology in the introduction and consistently using it throughout the results and discussion sections. This will help readers easily follow your references to homogenization or its opposite. Currently, similarity loss feels harder to grasp conceptually.

Here, you seem to suggest that frequent-species communities are more heterogeneous across distances within forested habitats, which is similar to managed grasslands. However, in grasslands, species communities across distances were more homogeneous compared to arable lands and settlements. Perhaps linking this observation to a specific figure or table would help clarify your point further?

L.161 to 170. Compared to the previous version, those results are better stated and described.

L.179. Figure 4 have greatly benefited from editorial work of the authors.

L.195. What does this imply? Does it mean that the pattern observed at the taxonomic level (BIN species) is preserved when considering body size classes (except for xxx)? In other words, is the distance decay pattern consistent regardless of whether the species are large or small? I would recommend simplifying the phrasing here to make your point as clear as crystal.

L.338. "distance decay depending on mobility". Please rephrase.

L.346: Could it be that species with low intrinsic mobility become passively mobile due to human presence? For instance, in settlements, species communities composed of both low-mobility and high-mobility species appear similarly heterogeneous. This could be explained by the lack of optimal habitats for highly mobile species to exhibit greater heterogeneity compared to other mobility categories, as you discuss. Alternatively, it could be that settlements facilitate passive transportation, increasing the mobility of low-mobility species.

L.383. You could refer to your nice Figure 1 for your study design.

L.397. Is 3x30 correct?

(Remarks on code availability)

Errors reported:

Either the loaded final csv matrix is false or the code needs to be updated as follow:

L.14 - 20 - 24 - 28: object name error

L.16 - 21 - 25 - 29: error in the number of column

L.57: needs to add install package for : library(snowfall)

L.66: where does the abline value comes from, I suggest you add a muted code corresponding to the calculation of this value as you did for Cmax_joint_f l.59.

L.82: needs to add install package for library(vegan)

L.91-92: for now with the data provided the matrices does not contain "samples" value required to run the function. Please modify accordingly.

Impossible to run the rest of the code. Please verify your provided data and accordingly modify the code so it can be run.

Version 2:

Reviewer comments:

Reviewer #2

(Remarks to the Author)

Review Nat-COMM-24-35277B

Reviewer Response Summary for Decker et al.

This study investigates how land use intensity drive arthropod diversity loss by examining both taxonomic and functional traits (body size and mobility) along a land-use intensity gradient across four land-use types and at two scales. Using extensive field sampling and advanced lab techniques, the researchers identify patterns of community homogenization. Contrary to expectations, the most homogenized communities were found in grasslands, while settlements and arable lands showed higher diversity between sites. Trait analysis revealed that larger, less mobile species had greater spatial heterogeneity.

Following reviewer feedback, the manuscript was significantly improved: the study's aims were clarified, trait-based expectations better contextualized, and the discussion made more cohesive. I value the authors' effort in ensuring terminological consistency across the text.

Aside from the brief remark below, I have no further comments to add.

Minor issues and line-by-line comments

L.93. Can you here already provide what are your four land-use types?

(Remarks on code availability)

Reviewer #3

(Remarks to the Author)

The manuscript entitled “Distance decay reveals contrasting effects of land-use types on arthropod community homogenisation” addresses the highly relevant and timely topic of biodiversity homogenisation driven by land-use change, based on a sophisticated and extensive dataset. The authors present a comprehensive study that contributes valuable insights to the field. I have carefully reviewed both the manuscript and the authors’ responses to previous reviewer comments. While many aspects have been adequately addressed, several methodological and structural issues remain and require further clarification or revision before the manuscript can be accepted.

Specific Comments:

- Line 449: Please provide the concentration of Proteinase K used in the extraction protocol to ensure full reproducibility.
- Lines 466–484: This section describing quality-filtering of the sequence data should be moved to the “Bioinformatics and Taxonomic Classification” section. Quality control and filtering are part of post-sequencing data processing and should logically follow the raw-data processing description.
- Line 486: Please provide the citation for the USEARCH software.
- Line 488: Cutadapt should be cited, and the version number used in this study should be included.
- Line 491: Please cite VSEARCH appropriately.
- Line 492: Why was filtering only based on minimum read length? It is standard practice, especially for COI metabarcoding, to apply both minimum and maximum length thresholds to remove nonspecific or chimeric fragments. Please justify this deviation.
- Pre-clustering rationale: The manuscript states that pre-clustering was performed prior to chimera filtering. However, de novo chimera filtering tools are typically efficient and fast. Please clarify the purpose and benefit of this step, as it is not clearly justified and may introduce unnecessary complexity.
- Line 498: The statement about using “BLAST” against the BOLD database is inaccurate. BOLD does not support direct BLAST-based queries in the same way as GenBank. Please clarify the exact procedure:
 - o When were the sequence similarity searches performed (I saw years in the read table)?
 - o Which BLAST algorithm was used (e.g., blastn, megablast)?
 - o Was a local database used or an online search?
 - o For GenBank, which database subset was queried?
 - o How was barcode identification against BOLD conducted - was this done via BOLDigger or directly via the website interface?
- Line 498 (read threshold): Please specify whether the 0.01% abundance threshold was applied globally across the entire dataset or per sample. This influences downstream diversity measures and interpretation.
- Line 500: The rationale for using the RDP’s Bayesian classifier after already conducting similarity searches against BOLD and GenBank is unclear.
 - o What was the added value of this second classification step?
 - o Which reference database was used with the RDP classifier?
 - o Overall, the taxonomic assignment workflow appears unnecessarily complex. Many studies, including comparable Malaise trap datasets, rely on a single step against BOLD. Please justify your multi-step approach and elaborate on its benefits over standard protocols.
- Line 502: BIN assignment is only possible via BOLD systems. Please explain how BINs were assigned in your workflow. Was a direct BOLD API or export used?
- Line 506: The term “dark taxa” must be clearly defined, contextualised, and supported by a reference. This term is not self-explanatory to all readers.
- Müller et al. (2023): The methodology for BIN alignment remains unclear. How were sequences aligned to BINs? Was a local copy of the BIN dataset downloaded and used for alignment? This step needs to be described in greater technical detail, especially if the method deviates from standard BOLD procedures.
- Line 520: Please revise the sentence to reflect that raw reads may be influenced not only by sequencing errors, but also by PCR errors, and primer bias.
- Figure 4: It appears that the diversity indices are mislabeled. Probably the plots were ment to be labeled as follows: $q = 0$; $q = 1$; $q = 2$.

Conclusion:

The authors have successfully addressed the majority of earlier reviewer comments and the study is, in principle, suitable for publication in Nature Communications. However, several important issues remain, particularly in the Methods section, regarding clarity, reproducibility, and methodological justification. Once these remaining concerns are adequately resolved, I would support the publication of this manuscript.

(Remarks on code availability)

Version 3:

Reviewer comments:

Reviewer #3

(Remarks to the Author)

The authors have adequately addressed all of my previous comments, and I have no further concerns. I can recommend the manuscript for publication in its current form.

(Remarks on code availability)

Reviewer #1 (Remarks to the Author):

Decker et al. explore the effect of land use on biotic homogenization in arthropod communities using a large metabarcoding dataset from Malaise traps in Bavaria. They sampled arthropod communities in four land cover types of differing land use intensity, forest, arable land, human settlements and grassland and explored the community's distance decay. When incorporating information on species abundances, as well as body size and mobility, they find that grasslands are the most homogeneous habitat type. Contrary to expectation, settlements and arable lands showed the spatially most heterogenous communities.

The work follows a very interesting and timely research question and I must commend the authors for the large amount of work put into this. While the manuscript is generally well written, the text contains various typos and grammatical mistakes. Unfortunately, I also see several issues with the study, which question the validity of the presented results.

We would like to thank the reviewer for the insightful comments, which significantly helped to improve the manuscript. Below is the point-to-point response to the comments and concerns.

My two main qualms are:

#1) Methodological details are lacking to judge the validity of the data: The study is based on a large DNA metabarcoding dataset and understanding this data is essential to judge the validity of the presented results. However, the methods lack all important details to judge on the reliability of the data. Considering that metabarcoding data is the core of this entire study, it is not sufficient to simply cite another study to provide the methodological approaches (the cited study also lacks many important details). The minimum information, which needs to be provided in the methods and results of the work are: What kind of field and lab controls were used and how were contaminations accounted for? What PCR and/or extraction replication strategy was used? What sequencing coverage was aimed for and were any libraries undersampled? The study cited as methodological reference refers to only a single Miseq V3 run. This would be by far not enough for a dataset as big as the one presented here.

Thank you for the suggestion, methodological details on metabarcoding can now be found in the methods section, in the "DNA sequencing" and "Bioinformatics" paragraph. The section includes DNA extraction and PCR details, controls, sequencing depth, and the bioinformatics pipeline:

L426-479 : “Next generation sequencing

Following the fractionating step, the preservative ethanol was removed, and the mixed arthropod samples were dried overnight in a 60–70°C oven to eliminate residual ethanol. The dried arthropods were then homogenized using stainless steel beads in a FastPrep 96 system (MP Biomedicals). DNA extraction from all samples was conducted by incubating them in a 90:10 solution of animal lysis buffer (buffer ATL, Qiagen DNEasy tissue kit, Qiagen, Hilden, Germany) with proteinase K. Following an overnight incubation at 56°C, samples were cooled to room temperature. DNA was subsequently extracted from 200 µL aliquots using the DNEasy blood & tissue kit (Qiagen) according to the manufacturer’s instructions. Amplicon PCRs employed 5 µL of extracted genomic DNA, Plant MyTAQ (Bioline, Luckenwalde, Germany), and HTS-adapted mini-barcode primers targeting the mitochondrial Cytochrome-Oxidase subunit I (CO1-5P) region (mlCOIntF/ dgHCO2198; see Leray et al., 2013; Morinière et al., 2016, 2019; Hausmann et al., 2021; Uhler et al., 2021). Amplification success and fragment length (313 bp) were confirmed via gel electrophoresis. Cleaned amplicons were resuspended in 50 µL molecular water. Illumina Nextera XT indices (Illumina Inc., San Diego, USA) were added in a secondary PCR using the same annealing temperature but limited to seven cycles. Ligation success was verified through gel electrophoresis, and DNA concentration measured with a Qubit fluorometer (Life Technologies, Carlsbad, USA). Samples were then pooled into 40 µL equimolar pools at 100 ng each, purified using MagSi-NGSprep Plus beads (Steinbrenner Laborsysteme GmbH, Wiesenbach, Germany), and eluted in a final volume of 20 µL. HTS was performed on multiple Illumina MiSeq (v3 chemistry, 2*300 bp, 600 cycles) runs with a target of 100,000 paired-end reads per sample. Sequencing depth was approximately 200k reads/sample. Six controls were included on each 96-well plate: 2 DNA-extraction control, 2 PCR control and 2 ligation control to account for contamination or false positive results.

Bioinformatics and Taxonomic classification

Paired-end reads were merged using the USEARCH suite’s -fastq_mergepairs utility (v11.0.667_i86linux32) with specific parameters (-fastq_maxdiffs 99, -fastq_pctid 75, -fastq_trunctail 0). Adapter sequences were trimmed using CUTADAPT K5, and untrimmed sequences were filtered with the --discard-untrimmed option. Further pre-processing, including quality filtering, dereplication, chimera filtering, and clustering, was handled through VSEARCH (v2.9.16). Initial quality filtering was implemented with --fastq_filter (parameters: --fastq_maxee 1, --minlen 300). Dereplication used --derep_fulllength (parameters: --sizeout, --relabel Uniq), conducted at the sample level and then combined, with

singletons excluded as noise. Following a 98% identity pre-clustering, chimera filtering was done using the centroids algorithm and VSEARCH's --uchime_denovo utility. Non-chimeric sequences were clustered into OTUs at 97% identity, with an OTU table generated through mapping reads to OTUs. False positives were minimized by excluding OTUs with read counts below 0.01% of total reads and by verifying OTU reliability against GenBank and BOLD databases via BLAST, following criteria for similarity and alignment quality. Taxonomic classification employed RDP's Bayesian classifier with COI dataset verification, and the resulting taxonomic summaries were visualized through KronaTools v1.311.

Taxonomic groups were then further resolved using the Barcode Index Number (BIN) system to define genetic units¹¹⁷, with identification supported by the BOLD platform (www.boldsystems.org)¹¹⁸. Single reads per BIN were removed, assuming sequencing errors. Due to gaps in existing libraries, assigning sequences to BIN units is challenging, especially for groups like dipterans, hymenopterans, and hemipterans that often include "dark taxa." For ecological analysis, we assigned sequences to units aiming to best represent species-level solutions and extracted ecological properties from sequence data. Following Müller et al. (2023), sequences were aligned with the nearest BIN in the study region, documenting genetic distance. BINs with <3% genetic distance were identified as BIN species, while those with >3% served as "genetic morpho-species" in ecological assessments. This process ensures that all sequences across lineages are balanced in taxonomic assignment, enabling reliable extraction of ecological properties like mobility and body size—traits often conserved at the genus or family level. “

#2) What was the taxonomic composition of the recovered samples, how comparable were the communities between habitat types? Why was no analysis performed for different taxonomic groups, for example orders? Considering the authors used Malaise traps, I assume there will be very dominant Diptera and Hymenoptera representation in the data.

Thank you for the comment. We made sub-responses to each issue: *#2a) the taxonomic composition of the recovered samples; #2b) how comparable were the communities between habitat types; #2c) why was no analysis performed for different taxonomic groups; #2d) Malaise traps.* We address *#2d) Malaise traps* in a separate issue, in issue #7.

#2a) the taxonomic composition: Taxonomic groups on the order level averaged per study plot (n=179) can now be found in the supplementary material, Supplementary Figure 1. This graph shows that orders are well represented in the samples from the four land use types. Diptera and Hymenoptera are the represented with the highest BIN numbers, but not overly dominating our samples when we look at average frequencies per plot. The mentioned figure can be seen here:

Supplementary Figure.1. The graphs show the average frequencies per study plot (a), and the total frequencies (b) of recovered BIN species across taxonomic orders (y-axes on a log-scale). Colours within bars indicate the proportions of total frequencies in the four land-use types (forest=green, grassland=purple, arable land = golden, settlement=grey). The pie chart (c) shows the total reads of BIN species across 10 taxonomic orders with the highest number of reads.

#2b) how comparable were the communities between habitat types: Our analysis ensures that the samples are comparable with the inclusion of sample coverage-based techniques (the framework of iNEXT by our co-author, Anne Chao) accounting for sample completeness. Comparing β -diversity measures across samples are more complicated than single-assemblage comparisons, because β -diversity depends on both among-sample differences and sampling effort or completeness. That is why we incorporated sample coverage-based analysis, and this way all samples are assessed on the same level of completeness, making samples comparable, regardless of which habitat type they were collected in. We included some more details in our methods section:

L123-135 (introduction): "To consider occurrence frequency, and the inherent incompleteness of insects samples, we applied a framework of community similarity along the Hill numbers⁵⁶ standardized by sample coverage^{57,58}. Rare and infrequent species often drive observed diversity patterns, therefore we accounted for changes in communities focusing on infrequent, frequent and highly frequent species separately⁵⁹. Given that distance-decay relationship is a form of β -diversity, it is based on α -, and γ -diversity, therefore this metric strongly depends on sample size, sampling effort, and sample coverage⁶⁰ (for more details of α -, β -, γ -diversity metrics, see Supplementary Info. Box 1). As a result, when estimating diversity and assemblage similarity metrics, it is important for assemblages (samples) to be statistically comparable across all study sites, which requires a standardised sample coverage-based analysis⁶¹. β -diversity values can be falsely modified when using unstandardised datasets: incomplete samples increase β -diversity values artificially⁶². Therefore, we adapted coverage-standardized indices of β -diversity to incidence data.";

and

L514-526 (methods): "Measures of β -diversity (and distance-decay) is calculated by gamma-diversity divided by alpha-diversity, which means that this metric depends on between-sample differences and therefore dependent on sample coverage. However, it is shown that most samples are incomplete, thus standardising sample coverage becomes very important^{60,133}. 'Sample coverage' was originally developed by Alan Turing in his cryptographic work, which is defined as the proportion of the total number of individuals in the entire assemblage that belong to the detected species in a sample. Chao et al.¹³⁴ extended Turing's concept of sample coverage to its incidence version to quantify sample completeness of replicated incidence data. This extension is based on the distribution of incidence frequencies obtained from multiple sampling units (such as DNA samples with identified BIN species). In our study,

sample coverage may differ between local land-use types, therefore we used coverage-based indices calculated by the package *iNEXT.beta3D*⁵⁸, which enables us to compare samples from different land-use types without any detection bias in our samples.”

#2c) why was no analysis performed for different taxonomic groups: Our study is focusing on community homogenisation, which can be best measured using all arthropod species data, instead of focusing on specific taxonomic groups. However, this creates very interesting insights of communities, focusing on specific groups would only identify the homogenisation rates within those specific communities, which was not the aim of the present study. Our studied land-use types most likely favour different sets of species, hence, to investigate the overall community homogenisation (or differentiation), we used all arthropod species data. In order to investigate fine-scale responses, we used species rarities based on Hill-numbers (discussed later) and trait data. This allowed us to discover broad patterns and draw general conclusions about arthropod community-level responses to land-use types, while further understand the overall state of arthropod communities. We added a sentence to the introduction to clarify this:

L63-70: “An improved understanding of the influence of land-use on β -diversity of arthropods is critical not only to maintain biodiversity, but also from a functional perspective¹⁴. Homogenisation often degrades ecosystem functions via losing sets of species from the community¹⁵⁻¹⁷, and arthropods, especially insects play crucial roles in ecosystem functioning¹⁸⁻²¹. Despite their high diversity²², there is still an ongoing debate on insect declines due to human activities²³⁻²⁵. An open research question is about the impact of land-use on insect whole-of-community responses. Our dataset contributes to the understanding of this question by using all-arthropod data and discover broad community-level patterns in response to land-use.”

#3) The ecological annotation, which is core to this work, is not reliable: The results are based on a classification of the recovered OTUs into abundance, body size and dispersal categories. The validity of these classifications is essential for this work. Abundance is identified based on repeatability of detection in 8 consecutive trapping events over 4 months. Using repeatability of occurrence is a great idea, but has not been properly tested in arthropods. But the main issue here is that the authors do not identify abundant species with their approach, but species with a temporally stable occurrence. Arthropod communities can show pronounced temporal community turnover in 4 months. Provided the sequencing was deep enough, the authors will detect even very rare species in each Malaise trap. Using the

author's approach, rare species with stable occurrence over time will be wrongly assigned as abundant. At the same time, highly abundant species, with a short seasonal occurrence would be judged as rare. You would need replicate traps at the same time point to apply the approach used by the authors here. I also feel that assigning body size for entire families is an oversimplification.

Thank you for the comment. We now use “infrequent”, “frequent”, and “highly frequent” in our manuscript, to better reflect the rarity of species.

Unfortunately, up to now, bulk-metabarcoding provides no true abundances, and the number of reads is a mixture of species biomass and number of individuals. However, statistical approaches have been developed using incidences instead of abundances in biodiversity estimates as shown in Colwell et al, 2012. This approach has been expanded recently to Hill numbers and to beta-diversity (Chao et al 2023). However, so far functions in statistical computing were not available for ecologists to easily transform incidence data to a raw matrix to which these methods for incidences can be applied to community distance matrices.

Therefore, we added such a function to the package *iNext.3D* which opens a broad avenue for all community analyses with incidence data rather than true abundances (e.g., sound recordings, camera trap data, replicated recordings by observers in the field, meta-barcoding). The new approach is mathematically based on theoretical literature and fits perfect to our data.

Following the framework of Hill-numbers, we assess communities focusing on *i*) infrequent (based on species richness, $q=0$), *ii*) frequent (based on Shannon diversity, $q=1$), and *iii*) highly frequent (based on Simpson diversity, $q=2$) species, not “abundant” species. This technique is not only useful for abundance-based datasets, but also is proven to reliably work with incidence-based datasets (Chao, A. et al., 2023 “*Rarefaction and extrapolation with beta diversity under a framework of Hill numbers: The iNEXT.beta3D standardization*”, Ecological Monographs, 84 (45-67); Chao, A., Hsieh, T., Chazdon, R. L., Colwell, R. K. & Gotelli, N. J., 2015. “*Unveiling the species-rank abundance distribution by generalizing the Good-Turing sample coverage theory.*” Ecology 96 (1189-1201)).

Habitat types (the four land-use types) were replicated at the same study period by having 55 forest, 45 grassland, 44 arable land and 35 settlement plots, and from each sampling period, every sample was fractionated to two sub-samples using a sieve with 8 mm mesh size.

Therefore, besides the temporal replication (8 time points), habitat types were also replicated, and each sample was divided into two subsamples before further DNA processing: for

example, a forest habitat type will have 55 plots * 8 sampling period * 2 subsamples = 880 replicates; or 55 * 2 = 110 replicates if we don't consider temporal replication. We believe that this level of replication combined with our sample coverage-based metrics (mentioned above, #2b, L518-530) should be enough to confidently represent arthropod communities from every sample.

We included some clarification in the text in L484-502: “We used frequency instead of raw reads from the sequenced data, which has issues due to sequencing errors¹²⁸. This method should also control for the fact that arthropod populations can be highly patchy in distribution both temporally and spatially, therefore abundance data could lead to species being over-, or underestimated in each sample community. It has been shown that replicated incidence data support statistical approaches to diversity estimation that are just as powerful as corresponding abundance-based approaches^{129,130}. Incidence data can be substituted for raw abundance data when the true number of individuals is not available¹²⁹, such as data derived from metabarcoding. While our frequency approach cannot estimate species aggregations¹³¹, it does detect spatial and temporal community aspects better than abundance-based techniques¹²⁹. In the current study, we distinguish between infrequent, frequent, and highly frequent BIN species within each community of the four land-use types (see below in section “Rarity”). Nevertheless, it is important to note that frequency data represent the presence of species, not the exact number of species per sample, however this should not change the overall patterns of whole-of-community responses. Frequency data ranged between 0 and 16, as we had samples from 8 collection events from two subsamples based on body size (described in sample collection). Besides the temporal replication (n=8), and the replicates of each habitat type, the inclusion of both sample fractions (‘big’ and ‘small’) introduces more replicates from the same sampling period.”

Please find further comments below:

#4) The authors refer to species throughout the text, for example, the abstract refers to 12,000 species. I feel this is misleading. They use 3 % radius OTUs for their work. OTUs can overestimate the true species diversity by severalfold, this will also happen with BOLD BINs. It would be better to refer to OTUs in the text.

Thank you for the suggestion, we changed the text and refer to ‘BIN-species’ in our text instead of species where applicable. We do not use OTU divergence, but species, or interim species, such as the Barcode Index Number for biodiversity assessments. It is true, that the

BIN algorithm (Ratnasingham et al., 2013 -

<https://journals.plos.org/plosone/article?id=10.1371/journal.pone.0066213>) uses an approximate 97% interval to cluster sequences into an interim species cluster, however the BIN system allows us to perform biodiversity estimates for species (or haplotype clusters, interim species) if they are not yet equipped with a full linnean taxonomy. Using BINs with >95%, allows us to differentiate between different haplotype clusters of CO1, which we then name BIN species, which provide a close match with species, as shown by Buchner et al. (2024): Buchner, D., Sinclair, J. S., Ayasse, M., Beermann, A. J., Buse, J., Dziock, F., Enss, J., Frenzel, M., Hörren, T., Li, Y., Monaghan, M. T., Morkel, C., Müller, J., Pauls, S. U., Richter, R., Scharnweber, T., Sorg, M., Stoll, S., Twietmeyer, S., ... Leese, F. (2024). *Upscaling biodiversity monitoring: Metabarcoding estimates 31,846 insect species from Malaise traps across Germany*. *Molecular Ecology Resources*, 00, e14023. <https://doi.org/10.1111/1755-0998.14023>

#5) The metabarcoding methodology is very critical to judge the validity of the work. I do not feel it is sufficient to cite Uhler et al. here. Uhler et al says their data was sequenced on one MiSeq v3 kit. This would be way too little data for a dataset of so many Malaise traps, as you present here. Recent work suggest sampling to several 100,000 to even millions of reads for a proper recovery of taxa in a Malaise trap.

Thank you, we added the missing information in the Methods section to “Next generation sequencing”. Samples were analyzed on multiple MiSEQ runs – sequencing depth per sample was approximately 200k reads/sample (100k paired end before bioinformatics).

#6) I am not sure is this study is based on data from a previous study (Uhler et al.) or if the data was generated for this work exclusively? If so, mention clearly in the methods

Thank you for pointing this out. The confusing sentences were omitted from the text, and a sentence was added for clarification:

L411-412: “Samples were collected, and a limited number of samples were processed already by Uhler et al., 2021. However, for the current study, the full collection was processed.”

#7) L362: “Nevertheless, this method reliably represents most groups of terrestrial arthropods, including flying and ground-dwelling group”

I beg to differ here. Malaise traps are very biased towards Diptera and Hymenoptera. For example, most Malaise traps are unsuited to sample beetles. Malaise traps are so popular,

because 1) insect decline was shown with them and 2) because they sample hyperdiverse Hymenoptera and Diptera groups, hence a very large species diversity. But not necessarily an exhaustive and representative spectrum of a community.

Malaise traps cover both flying and flightless insects, and they do not differentiate between different insect orders (as shown by Figure 1 in the Supplementary Material). This graph shows that most arthropod orders are well represented in our dataset, including Coleoptera. Interestingly, many flightless and predominantly litter and soil dwelling orders were recovered in our Malaise trap samples, such as order Dermaptera, Glomerida, Isopoda, Julida, or Lithobiomorpha.

Unfortunately, there is no one best method for insect trapping, given their amazing diversity and various lifestyle. We acknowledge that the most representative method would be to use Pitfall, Window, Malaise, and Light traps simultaneously, however this is not feasible for many studies, especially not for a large-scale design. However, with the results of another study currently under review we can underline, that Malaise trap is the most comprehensive method. In this study we explicitly compared Malaise traps, Pitfall traps and Window traps at the same sites. This data show that 1) Malaise traps alone collect by far the most species, and 2) the composition of species in the Malaise traps is between pitfall and window traps as it collects both flying, jumping, and climbing species.

a.

1) The above figure shows species α -diversity recovered from Pitfall (yellow), Flight interception (dark green), Malaise (light green) traps and the combinations of those: Pitfall x Flight interception traps (pink), Pitfall x Malaise traps (black), Flight interception x Malaise traps (brown); and the combination of all traps (blue).

2) The above figure shows the communities of the recovered insect orders trapped in different forests. The following orders are shown with Non-metric Multidimensional Scaling: Coleoptera (red dot), Diptera (green triangle), Hymenoptera (blue rectangle), Lepidoptera (blue cross), and all other orders (purple rectangle).

Some clarification was inserted to the text, L412-420:” As with any sampling method, using only Malaise traps for arthropod sampling introduces a bias in sampled taxa and not capturing all arthropod taxa equally¹¹³. Nevertheless, this method reliably represents most groups of terrestrial arthropods¹¹⁴, including flying and flightless groups (also shown in Supplementary Fig.1): ground-foraging arthropods climb up on the net, and some small insects can be carried by the wind where the net intercepts this passive flight. Even though these traps are less effective in catching large flying insects with improved vision (such as dragonflies), they are still the most efficient trapping method for mass-collection of arthropods^{115,116}.”

#8) L369 what is COI 5P? I assume it refers to the 5-prime region of the COI gene. You could just say the COI barcode region.

Cytochrome-Oxidase subunit I is correct. Now this is in the main text.

#9) L380 – 381 I assume the first one is supposed to be < 3 %

Thank you, corrected.

#10) L386 repeating the same info twice

Thank you, this was deleted and replaced with the details of the analysis.

#11) L392: change to “abundance data”

Thank you, corrected.

#12) L395: “By using frequency, we consider species captured at all sampling periods (captured 16 out of 16 times) as species with high relative abundance”

As stated above, this is not true. See my explanation above. You would need local subsampling, e.g. replicate traps from the same time window. As your entire study is based on this assignment, this is a big problem.

Thank you, we added some further clarification and replaced relative abundance, as the use of this term here was not correct. We study community compositions between assemblages; therefore, our analysis is valid using frequency data. Incidence frequency is a robust metric to serve as a proxy for species abundance in each sample, as shown in Colwell et al, 2012.

L481-502: “We used detection frequency (or incidence) in our dataset. Our data consisted of samples from 179 sites, with 8 sequential repetitions. Then, each sample was divided into two fractions, depending on body size. This means that each BIN-species had a frequency value (the count they occurred) ranging from 0 to 16 for all our 179 study sites. We used frequency instead of raw reads from the sequenced data, which has issues due to sequencing errors¹²⁸. This method should also control for the fact that arthropod populations can be highly patchy in distribution both temporally and spatially, therefore abundance data could lead to species being over-, or underestimated in each sample community. It has been shown that replicated incidence data support statistical approaches to diversity estimation that are just as powerful as corresponding abundance-based approaches^{129,130}. Incidence data can be substituted for raw abundance data when the true number of individuals is not available¹²⁹, such as data derived from metabarcoding. While our frequency approach cannot estimate species aggregations¹³¹, it does detect spatial and temporal community aspects better than abundance-based techniques¹²⁹. In the current study, we distinguish between infrequent, frequent, and highly frequent BIN species within each community of the four land-use types (see below in section “Rarity”). Nevertheless, it is important to note that frequency data represent the presence of species, not the exact number of species per sample, however this should not change the overall patterns of whole-of-community responses. Frequency data ranged between 0 and 16, as we had samples from 8 collection events from two subsamples based on body size (described in sample collection). Besides the temporal replication (n=8), and the replicates of each habitat type, the inclusion of

both sample fractions ('big' and 'small') introduces more replicates from the same sampling period. “

#13) L430 traits: body size was assigned at family level. I feel this an oversimplification. This may work in some abundant families like Cecidomyiidae or in whole groups like Collembola or Acari. But many arthropod families show massive variation in bodysize. This is also true for the hyperdiverse Hymenoptera and Diptera dark taxa which dominate Malaise traps. It is true that family-level body size is a simplified measure, however it is a valid metric to discover general patterns and to classify species in coarse categories such as our body size class. An additional figure is now included in the Supplementary Material, Figure 4, where we tested the estimated values (used in the manuscript) against a subset of families with exact body size values for species (Gossner et al., 2015). The estimates for families are very close to the values which were measured on single species within the many families. Besides the close size value match, the categories we used in the manuscript (large, medium, small) allow more flexibility, because even when body size values are not the same, the individual category of a family should reflect reality more accurately.

L561-564: “To confirm the body size categories, a subset of measured body sizes of species belonging to 71 families (Martin M Gossner et al., 2015) were tested against our estimated body size categories, and revealed a close match between these families (Supp. I, Fig. 4).“

Supplementary Figure 4. The figure shows the relationship of body size estimates and average body sizes of 71 arthropod families based on species-specific data from Gossner et al., 2015. “A summary of eight traits of Coleoptera, Hemiptera, Orthoptera and Araneae, occurring in grasslands in Germany”, *Sci Data* 2, 150013. <https://doi.org/10.1038/sdata.2015.13>. Boxes indicate the size classes (green = small, red = medium, brick = large) used in the current paper based on estimates, and the red dashed line indicates the goodness of match between the two datasets (estimated and real sizes).

#14) Traits: Your dataset is probably very biased towards Hymenoptera and Diptera. How do the observed patterns change between different taxonomic groups? Why do such a taxonomically coarse analysis, if barcoding allows identification of all groups to fine scale taxonomic levels?

As mentioned above, the orders averaged over study plots show that there is no single order dominating all of our samples. In fact, Hymenopteran and Dipteran are the most diverse orders in our habitats. We aimed to provide a generalised pattern of overall arthropod communities in four land-use types. Our goal was not to determine the response of single arthropod groups separately, but to measure species community homogenisation (or heterogenization), which requires the full spectrum of taxa. To show land-use dependent community patterns, we used all recovered arthropod data, which gave us a whole-of-community response to land-use, instead of focusing on single taxa. To our knowledge, whole-of-community arthropod responses to land-use was not done before, therefore focusing on single taxonomic groups would compromise the novelty of our study.

#15) Also, was your trait datasets taxonomically biased? E.g. were certain taxa/orders overrepresented in the small or big or high or low dispersal category? Could taxonomy be a driver of the observed effects? How did you control for this?

Supplementary Information I. Fig.1 shows that our data is not taxonomically biased. As mentioned above, we work with a community dataset, and this is also true for the trait aspect of the study. Hill-numbers are very useful and robust when species are not responding to the studied variable the same way or if certain groups occur more than others. Using Hill-numbers, we can show that when certain taxa are “overrepresented” within a community, it will be weighed in the community focusing on highly frequent species ($q=2$), however this will be ignored when focusing on communities of infrequent species ($q=0$).

#16) L432: why do you have hexanauplia in your data? Aren't they only aquatic arthropods?
Thank you for the comment. This was unfortunately a mistake in the text, we did not consider aquatic organisms in our final dataset. The text should have included Diplura. This is now corrected.

#17) L478: Data availability: Metabarcoding data need to be available, that includes OTU tables as well as all raw reads.

Thank you for the comment. The used dataset is available on Figshare: incidence frequency data of recovered BIN species. We also uploaded the raw metabarcoding identifications and the corresponding fasta files for all OTUs to be fully transparent; however, this was not used in our analysis.

Reviewer #2 (Remarks to the Author):

Review Nat-COMM-24-352777-T

Review summary

The paper is overall well-executed, with a clear and compelling presentation of the research context. While the topic itself may not be entirely novel, the study distinguishes itself by applying innovative fieldwork techniques to test new hypotheses within the field. This adds significant value and relevance to the existing body of knowledge. The authors have effectively articulated the significance of their work, making it a valuable contribution to the ongoing research in this area.

In their study, the authors investigate the impact of landscape homogenization across distances on insect diversity loss by examining four distinct landscape types: forests, managed grasslands, arable lands, and settlements, all within near-natural, agricultural, and urban areas. They hypothesized that the decline in beta-diversity over space (distance-decay) would intensify with increased land-use intensity. Focusing on arthropod communities, however, they found results contrary to their expectations. The most homogenized communities were observed in grasslands, while the greatest diversity between locations was found in settlements and arable lands. Beyond community homogenization, they also explored species traits within these communities. As anticipated, larger, and less mobile species exhibited more spatial heterogeneity than smaller, more mobile species. The title of the submitted article highlights the complexity of these findings. Additionally, the article delves into these results, which contradict expectations and previously published literature, offering insight into the complexities of land use management and its effects on arthropod communities.

While the manuscript has significant strengths and addresses a highly interesting topic, it requires major revisions to enhance clarity and emphasize the main message. The finding that the most anthropogenic environments remain more heterogeneous than the 'most natural' ones is particularly unexpected, yet it is not sufficiently highlighted in the abstract or discussion. Given the complexity of the study, which spans various scales (both temporal and geographical), land-use types, species rarity, and traits, I recommend creating a schematic representation of the study to help readers follow the concepts more easily (an improved version of Figure 0). Additionally, I suggest revisiting some discussion points for better emphasis and possibly reorganizing the discussion to present the ideas in a more logical

sequence. Lastly, several minor revisions are needed to correct sentence errors and improve overall readability. All comments and suggestions are provided in detail in the following review.

We would like to thank the reviewer for the thorough review and the comments on this manuscript. The comments should improve the manuscript and the detailed responses are written below.

Major issues

#18) First, as outlined in the summary, the main messages regarding the complex diversity response to different land-use types are not clearly emphasized in your abstract and discussion. As a result, the reader may struggle to grasp the key takeaways. This may be due to the inherently complex structure of your study, which is indeed a strength. However, improving certain sentences and refining Figure 0 could significantly enhance the clarity and coherence of the main text. I suggest revising Figure 0 to align with the various steps of your study as described in the Methods section (i, ii, iii). This will help clarify the scales of analysis, which can be somewhat confusing—particularly when distinguishing between the local level (four types of environments) and the regional scale (near-natural, agricultural, urban). Additionally, some of the minor, line-by-line comments may further assist in addressing these points.

The main messages are described clearer in the Abstract L32-36: “. Our approach for incidence data under consideration of incomplete samples - taking frequency and species traits into account - identified that grasslands harbour the most homogenous communities. In contrast, the most modified land-use types, settlements, and arable lands did not differ from forests, and showed the most heterogeneous communities between locations.”;

and the Discussion was written clearer, starting with the main finding, L234-242: “Our research extends the study of Gossner et al¹⁰ from grasslands to other land-uses, which showed increasingly similar arthropod communities with increasing grassland management intensity. However, we did not find evidence for species homogenisation of communities along the increasing land-use gradient from forest, to managed grassland, to arable land and settlement. Instead, species communities were the most homogeneous in a less intensely managed land-use type: managed grasslands harboured the most similar arthropod species over spatial distances.

In contrary to our expectations, the two most modified land-use types, arable lands and settlements exhibited the most heterogeneous arthropod communities between locations.”

Thank you for the suggestion, we included more details to Figure 1 (previously Figure 0). Now the figure includes the analysis steps; the hypotheses and results on responses to land-use by arthropod communities and the responses focusing on communities with different traits.

Methods are now described in more detail. We clarified some methods in the introduction to emphasize the structure of the design in L100-106: “First, we tested distance decay-patterns at a local land-use scale, because arthropod assemblage variations are best explained by local habitat characteristics⁴³. Then, we assessed the outcome of these local subsets in three types of regional landscapes, which could modify local processes⁴⁴⁻⁴⁶. ‘Local land-use’ is considered the immediate surrounding of arthropod traps (0.5 ha), which impacts arthropod communities the most⁴⁷. ‘Regional landscape’ describes the broad area (5.8 km x 5.8 km quadrant), not only the realised habitat of an arthropod. “

#19) Overall, the text could be made clearer by more precisely defining terms used throughout. For example, please clarify whether 'strong distance decay' refers to 'high heterogeneity,' and then consistently use one term to avoid confusion. The same applies to the abbreviations for beta and alpha diversity—define them clearly and stick to one throughout the text.

We now included a Box to the Supplementary Information to explain the following terms: alpha, beta-, gamma-diversity; distance decay, community homogenisation or heterogenisation. L127-130: “Given that distance-decay relationship is a form of β -diversity, it is based on α -, and γ -diversity, therefore this metric strongly depends on sample size, sampling effort, and sample coverage⁶⁰ (for more details of α -, β -, γ -diversity metrics, see Supplementary Info. Box 1).”

#20) Additionally, the phrase 'on a scale' is frequently used; however, 'at a scale' may be more appropriate in most contexts. For instance, it's correct to say, 'on a scale from local to global,' but 'at the global scale' is more accurate.

Please also ensure consistency in the use of either American or British English throughout the manuscript. Furthermore, some sentences are quite long, making it difficult for readers to grasp the main points quickly. Examples of this are provided in the line-by-line comments.

Finally, I suggest making your Conclusions more concise. While all the necessary points are present, the lengthy sentences can obscure the main message, making the text and its justifications harder to follow.

We corrected some language mistakes and use British English throughout the text.

#21) Finally, a few additional points that could enrich the discussion are worth considering. These are detailed in the line-by-line comments, but for instance, there is limited discussion on forest composition, despite noting a potential bias in the methodology. Similarly, the grassy areas within settlements have been mentioned, but could benefit from further elaboration.

We added a paragraph on forest communities in L243-250: “The most natural local land-use type did not have the strongest distance-decay as expected, meaning that forests did not harbour largely different sets of species between locations. The arthropod communities with higher-than-expected homogeneity in forests are likely driven by current and historic Central European forest management. The mostly uniform management practices lead to even-aged stands with mostly closed canopies and lacking old-growth tree patches⁶³⁻⁶⁵. Consequently, even though forests had the highest α -diversity (meaning that this land-use type is the most locally diverse in arthropod species)⁶⁶ the created analogous habitats in forests lead to communities consisting of similar species between forest patches⁶⁷.”

Settlement mosaics are also mentioned in more detail in L278-282:” Differences in practices are also present in settlements, but at a smaller scale: gardens in settlements have a huge range of grown plants and types of garden maintenance. Thus, even though urban expansion is a major cause of habitat loss for many species^{85,86}, increasing attention is given to the beneficial impacts of dense habitat mosaics for species in settlements⁸⁷⁻⁸⁹.”

And L305-310: “Urban features add diverse hospitable habitats for otherwise forest-inhabiting arthropods, such as small hobby farms, private gardens, riparian corridors and even small remnant vegetation^{98,99}. Such contrasting habitat types could act as complementary or supplementary environments explained by the ‘cross-habitat spillover hypothesis’⁹²: species flow between land-use types depending on their temporal and spatial requirements. This could then enhance local species pools, and therefore increase β -diversity between forest locations.”

Minor issues and line-by-line comments

#22) l.1. *Prepared for Nature Ecology and Evolution?*

This was corrected, thank you.

#23) l.20. *Verify affiliation, Leipzig? Guess it is Leipzig*

This was corrected, thank you.

#24) l.22. *Please remove one repeated 'during'*

This was corrected, thank you.

#25) l.30-31 *Please rephrase, what is tested is unclear here*

The text has been rephrased in L29-32: "We studied communities along an increasing local land-use intensity gradient from forests to managed grasslands, to arable lands and to settlements situated within near-natural, agricultural, and urban regional landscapes."

#26) *Figure 0: I suggest adding directly on the figure that left distance decay represent Hypothesized distance decay whereas on the right it is the actual result of the paper.*

The figure (now Fig.1) is improved with more detailed results.

#27) l.50. *I disagree with the fact that specialized species decline in human-modified habitats. In fact, some very specialized species are selected. Some others are counter selected. Maybe you can be more specific here or modify your sentence. This is explicitly the title of the first article you cite.*

The sentence was modified in L52-55: "Excessive habitat modification drives species composition, favouring sets of species, dramatically changing their composition in natural habitats^{1,2}. Habitat modification includes intense land-use, which causes both biotic homogenisation and heterogenisation^{3,4} when communities can become more or less similar across space^{5,6}. "

#28)l.51. *More or less*

Corrected.

#29) l.52 and 54. *Beta-diversity, please define and use the same expression everywhere*

We included a Supplementary Box to explain the used terms of diversity metrics.

#30) l.62. *Maj is missing. Also stick to one term: homogenization or homogenisation?*

We corrected the use of homogenisation.

Now the sentence is broken down into two sentences.

L63-67: "An improved understanding of the influence of land-use on β -diversity of arthropods is critical not only to maintain biodiversity, but also from a functional perspective¹⁴. Homogenisation often degrades ecosystem functions via losing sets of species

from the community¹⁵⁻¹⁷, and arthropods, especially insects play crucial roles in ecosystem functioning¹⁸⁻²¹. “

#31) l.74. *Various different?*

This sentence was deleted.

#32) l.73-76. *Here I do not get the main idea you are talking about. Are you saying that studies at global scale cannot conclude on the factors that contribute to communities' changes? Are you saying that studies looking at the effect of management of land-use specifically are scarce? Please rephrase.*

Thank you, we rephrased the sentence in L78-89: “While global distance-decay studies contribute to the understanding of community responses to environmental changes²⁸, using intercontinental datasets could be problematic. It is widely shown that besides spatial scale²⁹⁻³¹, distance-decay also depends on study extent and design³². Regional and local diversity measures are used in conservation planning⁹, but study designs that offer available data at the appropriate scale are often lacking³³. Moreover, community change patterns are more likely to be related to complex environmental and climatic differences when data is obtained across ecoregions, covering great geographic distances^{11,28,34}. In contrast, management-relevant studies seek to find similar ecosystems, which allows for comparisons of habitats shaped by human impacts instead of large-scale environmental drivers. Comparative studies between land-use types, but within similar ecosystems are relatively scarce apart from some exceptions³⁵⁻³⁸. “

#33) l.78. *Please add a link with previous paragraph.*

The text in these paragraphs have been modified and moved.

#34) l.80. *What are Malaise traps? Maybe cite a research work on that.*

Malaise traps are described in the Methods: Sample collection section.

L407-411: “Malaise traps were used to capture invertebrates with the following dimensions: height front: 0.90 m; height rear: 0.60 m; length: 1.60 m, with 80% ethanol as preserving solution. Malaise traps are tent-like structures, which captures arthropods by intersecting their ground movements or flights. Sampling containers are on top of the “tent”, using arthropods tendency to move upwards¹¹⁹. “

#35) l.81. What do you mean by macro-environmental differences? What were you looking at specifically?

This text has been modified in L81-87: “Regional and local diversity measures are used in conservation planning⁹, but study designs that offer available data at the appropriate scale are often lacking³³. Moreover, community change patterns are more likely to be related to complex environmental and climatic differences when data is obtained across ecoregions, covering great geographic distances^{11,28,34}. In contrast, management-relevant studies seek to find similar ecosystems, which allows for comparisons of habitats shaped by human impacts instead of large-scale environmental drivers. “, and then in L91-94: “The design provides a standardised dataset where biogeographic patterns play little role in distance-decay relationships. Instead, arthropod community patterns are driven by land-use intensity within the same ecoregion³⁹.”

#36) l.82 on?

We now only use “at a ... scale” instead of also using “on a scale”. The sentence now reads as follows: “We tested distance decay patterns at a local scale, where arthropods are impacted by their immediate surroundings...”

#37) l.82-84. Unclear what is the local scale, what is the regional scale? Some precision might be needed for a better understanding.

Some details have been added to the introduction. The text specifies the scales in L100-106: “First, we tested distance decay-patterns at a local land-use scale, because arthropod assemblage variations are best explained by local habitat characteristics⁴³. Then, we assessed the outcome of these local subsets in three types of regional landscapes, which could modify local processes⁴⁴⁻⁴⁶. ‘Local land-use’ is considered the immediate surrounding of arthropod traps (0.5 ha), which impacts arthropod communities the most⁴⁷. ‘Regional landscape’ describes the broad area (5.8 km x 5.8 km quadrant), not only the realised habitat of an arthropod. “

#38) l.82-86. Very unclear here, it is a bit confusing to talk about local scales/regional scales and then local land-use types. Can you please rephrase?

The text specifies the scales in L100-106: “First, we tested distance decay-patterns at a local land-use scale, because arthropod assemblage variations are best explained by local habitat characteristics⁴³. Then, we assessed the outcome of these local subsets in three types of regional landscapes, which could modify local processes⁴⁴⁻⁴⁶. ‘Local land-use’ is considered

the immediate surrounding of arthropod traps (0.5 ha), which impacts arthropod communities the most⁴⁷. ‘Regional landscape’ describes the broad area (5.8 km x 5.8 km quadrant), not only the realised habitat of an arthropod. “

#39) *Figure 1b. Can you change the map font, for more visibility?*

We assume that the ‘font size’ on Fig. 1b (now Fig. 2b) refers to the regional study site symbols, because there is no text on Fig. 1b (now Fig. 2b). The colours have been modified to make the symbols of regional landscape types more distinguishable.

#40) *l.109. What are time periods? Until then it is nowhere mentioned?*

Time periods are mentioned in the Methods section in L420-422: “Traps were active between April or May and August 2019, with a fortnightly collection for 4 months, resulting in 8 sampling periods in total. Timing of sample collections varied due to weather and snow cover differences between locations.”

#41) *l.110. What is BINs the acronym for?*

BINs are now mentioned in the text in the introduction for clarity, L98: “We used arthropod assemblages of ~12k BIN species (Barcode Index Numbers, see methods) “, and then in the methods, L471-473: “Taxonomic groups were then further resolved using the Barcode Index Number (BIN) system to define genetic units¹²⁶, with identification supported by the BOLD platform (www.boldsystems.org)¹²⁷.”

#42) *l.114 to 116. This is an interesting result around species richness that can be discussed.*

This result on α -diversity is in accordance with the previous study on arthropods using the same study design, Uhler, J. *et al. Relationship of insect biomass and richness with land use along a climate gradient*. *Nature communications* **12**, 5946.

Therefore, in our subsequent study, we focus on the β -diversity component (distance-decay). We discuss this briefly in the text in L280-284:” Thus, even though urban expansion is a major cause of habitat loss for many species^{85,86}, increasing attention is given to the beneficial impacts of dense habitat mosaics for species in settlements⁸⁷⁻⁸⁹. Similarly to our finding for β -diversity, Uhler *et al.*⁹⁰ reported high species richness of arthropods in settlements - despite low biomass -, especially within near-natural regions.”;

and L248-250:” Consequently, even though forests had the highest α -diversity (meaning that this land-use type is the most locally diverse in arthropod species)⁶⁶ the created analogous habitats in forests lead to communities consisting of similar species between forest patches⁶⁷”.

#43) Figure 2. Why are the index value to calculate distance decay different when looking at different species type? Please help the reader (although explained in Methods).

This comment is not too clear for the authors. There are no species types (assuming trait categories) shown on Figure 2 (now Figure 3). The index value is always the steepness of the slope which is derived from the distance-decay curve (Figure 3a: species similarity distance X physical distance). This is the same for all species types (trait categories) in all land-use types.

#44) l.154. Not significantly different from grasslands that also records small species on average.

The difference is not significant, but the value is the smallest in settlements. Significance is now included in the sentence in L188-193:” On average, forests harboured the largest arthropods (6.21 ± 0.05 mm), but not significantly differing from those in arable lands. The smallest species occurred in settlements (5.85 ± 0.07 mm), but not significantly different from those in grasslands and arable lands. Species in arable lands had the highest mobility score (2.72 ± 0.003), significantly higher than those in forests and grasslands; while the other three land-use types were similar in their mobilities (Fig. 5, Supplementary I. Table 2)”.

#45) l.155. Maybe this is interesting to note that arable land might be heterogeneous because they record the most mobile species among their communities?

Thank you for the suggestion. This pattern would be quite difficult to interpret because the highly mobile species showed the weakest distance-decay. They are mobile enough to travel between patches, covering larger areas than the less-mobile species. This makes the communities more similar to each other, because the same high-mobile species are occurring in most sites, and at sites which have large distances between them.

#46) l.157. Even if we may know that species size has something to do with species life history traits, dispersal, or mobility, I think there is a need to explain why small or large bodied species are expected to behave differently, as finally you do not find any difference (l.157).

Additionally, I find the paragraph on body size and homogenization in different land use type a bit hard to read, is not there another way to say the things? Large species are more homogeneous in settlements, arable lands, and forest for typical species. Additionally, nothing is discussed here for grasslands.

We expanded our hypothesis on large arthropods in L120-122: “Large-bodied arthropods require bigger patches of optimal habitat⁴⁵, and often favour low disturbance regimes^{46,47}, most likely due to the limited ability to escape”; and

L312-315: “Large-bodied arthropods require larger patches of suitable habitat^{53,100}, and low disturbance regimes^{54,55,101}. This is reflected in our results, where forests harboured communities with the largest arthropods, while the smallest ones were found in the most modified land-use types, in settlements.”

In the results, we tried to simplify the text and used ‘higher or lower community similarity loss with distance. L194-198: “Body size did not significantly influence community similarity loss with distance in most land-use types. Significant differences were only detected in communities focusing on highly frequent species ($q=2$). In forests, small arthropods had a higher community similarity loss with distance than medium arthropods. In arable lands, medium arthropods showed significantly the highest community similarity loss with distance. In settlements, large arthropods had higher community similarity loss than small ones. (Fig. 6, Supplementary I. Table 3). “

#47) Figure 5. No change in communities’ composition for rare species whatever size and habitat. It could be nice to add something pondered by the richness in the different land-use type? Maybe it could be nice to help the reader understand the figure to add an axis how to read heterogeneity and homogeneity along the distance decay axes.

Now Figure 6. We inserted a sentence to address the lack of rare species responses depending on body size in L333-336: “Body size only affected distance-decay when focusing on highly frequent species, which could be a spatial extrapolation to the findings of van Klink et al., where dominant species had the strongest responses to environmental changes over time²⁴”.

To make the x-axes more straightforward, we changed the x-axes to “Community similarity loss with distance” on figures 3, 6, 7 (previously figures 2,5,6).

#48) L.184. Less community similarity: maybe define appropriately homogeneity and heterogeneity once and then stick to that term. Sometimes you say more heterogeneity, sometimes less similarity. Maybe on that figure you can also highlight that grasslands are

very often more homogeneous than the other land-use type. I recommend revising this figure to simplify its interpretation and clearly convey the key messages.

A Box in the Supplementary Information is now included to explain the terms linking similarity with homogenisation. These terms are now used less interchangeably in the text, however, to avoid frequent repetition, we still use both terms.

The result on land-use types highlighting that grasslands are the most homogeneous can be found on Figure 3. The result and this figure have been updated.

#49) l.169. Something here interesting on the fact that intermediate mobility species seems more heterogeneous in community than low and high.

This result has been updated.

#50) Figure 6. From this figure, I would primarily conclude that insects with low mobility differ between locations with grassland and forest land use types, which is quite interesting. It appears that the dispersal pattern is only evident in the more “natural” environments, such as grasslands and forests, and not in arable lands or settlements. This observation could be explored in relation to exotic species, species dispersal abilities, and species richness at these locations.

Now Figure 7. The results and figure has been updated. Species dispersal abilities are investigated by using communities only harbouring low-, intermediate-, and high-mobility species. The found pattern is discussed in L337-344: “In contrast to body size, distance-decay depending on mobility was affected by local land-use and species rarity. Although the patterns were not uniform, the disadvantage of low mobility species in frequently disturbed land-use types^{51,111} was clearly reflected in our results. Communities including low-mobility species were the most heterogeneous in arable lands and grasslands. Highly mobile species can escape the disturbance and rapidly recolonizing after, while low-mobility species must find local refugia, therefore populations might become very patchy, increasing heterogeneity between locations, and strengthening distance decay.”

#51) l.197. What is the difference between this lines and l.76? Same sentence, same papers. Additionally, this citation style is somewhat awkward. & l.199. see is not needed.

The discussion starts with highlighting (or repeating) the reason we did this study and the shortcomings of global meta-analysis of this topic. However, we deleted this section to avoid

repetition and highlight our major findings earlier. We cannot see why exactly the citation style is awkward.

#52) l.196 – 200. A lot of “but”. Please rephrase this paragraph to be more direct and break it into separate sentences. &

This has been corrected.

#53) l.201-202. “Regional and local diversity assist in conservation planning”? “designs of available data often do not allow for testing » could you please rephrase in something more direct? Something like: Conservation planning is based on regional and local diversity. However, study designs that offer available data for testing diversity at the appropriate scale are often lacking.

The mentioned section was improved and it is now in the introduction in L78-89: “While global distance-decay studies contribute to the understanding of community responses to environmental changes²⁸, using intercontinental datasets could be problematic. It is widely shown that besides spatial scale²⁹⁻³¹, distance-decay also depends on study extent and design³². Regional and local diversity measures are used in conservation planning⁹, but study designs that offer available data at the appropriate scale are often lacking³³. Moreover, community change patterns are more likely to be related to complex environmental and climatic differences when data is obtained across ecoregions, covering great geographic distances^{11,28,34}. In contrast, management-relevant studies seek to find similar ecosystems, which allows for comparisons of habitats shaped by human impacts instead of large-scale environmental drivers. Comparative studies between land-use types, but within similar ecosystems are relatively scarce apart from some exceptions³⁵⁻³⁸.”

#54) l.204. Can you improve the phrasing?

This is now in the introduction, L91-94: “The design provides a standardised dataset where biogeographic patterns play little role in distance-decay relationships. Instead, arthropod community patterns are driven by land-use intensity within the same ecoregion³⁹. “

#55) l.207. No need for “see also”

Noted, thank you.

#56) 1.215. *Some discussion might be needed on that point. When looking at mobility, low mobile species are not more heterogeneous between locations in land use type arable and settlements whereas there is a difference in grasslands. So maybe something has happened in those areas regarding low mobile insects? What I mean here, is that with distance between location, you may expect low mobile species to differ more than highly mobile species.*

This is the result we can see here: low-mobility species have a steeper distance-decay curve, and they differ more spatially than communities focusing on high-mobility species. We mention this pattern in the discussion, L337-344: “In contrast to body size, distance-decay depending on mobility was affected by local land-use and species rarity. Although the patterns were not uniform, the disadvantage of low mobility species in frequently disturbed land-use types^{51,111} was clearly reflected in our results. Communities including low-mobility species were the most heterogeneous in arable lands and grasslands. Highly mobile species can escape the disturbance and rapidly recolonizing after, while low-mobility species must find local refugia, therefore populations might become very patchy, increasing heterogeneity between locations, and strengthening distance decay. ”

#57) 1.228-229. *You might also discuss how agricultural practices and pesticide use significantly contribute to the reduction of various insect species in certain arable regions, while these practices may differ substantially in other areas. This could explain or support your findings of greater heterogeneity in areas with more intensive land use.*

We included some interpretation on this process, thanks for the suggestion. L271-280:” For example, a couple of species could specialise on a given crop in one arable land, while on another arable land a different plant is grown with different sets of highly specialised (or pest) species. This way, only a few number of high-occupancy species benefit from each arable land patch, decreasing α -, but increasing β -diversity³. Besides crop heterogeneity, differences in agricultural practices could create complexity, which selects for different species sets at different arable lands. For example, different types of intercrop ground cover will change the species which could overwinter at a given arable land; or the method of fertilisation (timing, type of organic input, etc.) can also lead to different species compositions⁸⁵. Differences in practices are also present in settlements, but at a smaller scale: gardens in settlements have a huge range of grown plants and types of garden maintenance.”

#58) l.232. *What is reflected in your study? You are talking about arable field size. I guess you are mentioning the fact that the intensity of arable land use is less important than in Germany. Please rephrase and justify. The justification follows subsequently. I suggest something like: In accordance with recent studies, our results confirm the importance of high crop heterogeneity and field size on biodiversity.*

Corrected, thank you. L265-267: “In accordance with recent studies, our results also confirm the importance of high crop heterogeneity and small crop field size, which can be beneficial to biodiversity, even to a greater extent than natural land cover^{75,76}.”

#59) L.235-239. *Well, you never mention what is alpha diversity in your article. So please explain it somewhere or rephrase. It is hard to understand how you can compare alpha diversity (within location diversity) and your distance decay based on beta diversity (between location diversity). Please explain and rephrase this paragraph.*

We added a Box to the Supplementary Information explaining the diversity metrics. β -diversity = $\alpha \div \gamma$ -diversity.

#60) l.242. *alpha-diversity of arthropods species richness, seems that you repeat or misuse the terms. Alpha diversity is already a measure of the species diversity OR richness.*

We changed this sentence to improve clarity in L282-284: “Similarly to our finding for β -diversity, Uhler et al.⁵¹ reported high species richness in local assemblages of arthropods in settlements, - despite low biomass - especially in settlements within near-natural regions.”

Alpha-diversity refers to the scale we measure diversity, referring to within-assemblage (local) species diversity. Richness is the raw number (count) of species present in an assemblage, while other metrics, such as Shannon-index or Chao1, estimate diversity based on ‘unseen’ species, species frequencies or abundances (rather than simply counting species).

#61) l.249-252. *Interesting part of the discussion, but it is not what you observe for grasslands except if I misunderstood?*

This refers to the regional result: when grasslands are surrounded by near-natural regional landscapes, communities become more heterogeneous with increasing distance (Fig. 4). The sentence was improved for clarity in L289-292: “When the more intense land-use types are surrounded by near-natural landscapes, species can move between these areas. Species can

escape disturbance periods, such as harvesting on arable lands and mowing on grasslands to the surrounding near-natural areas and recolonise from there⁹³”.

#62) l.254-255. *passive sentence please rephrase.*

L294-295: “At the same time, being surrounded by agricultural areas further increased species homogenisation at the more natural land-use types: forest and grassland.”

#63) l.258. *Do you mean forests isolated by urban environments? If yes, please rephrase.*

L299-300: “This heterogenising effect could imply that forests are isolated by urban environments⁹⁵. ...”

#64) l.317. *“to four”*

Corrected.

#65) l.320. *Now is gamma diversity that has not been defined earlier ... I suggest you define beta, alpha and gamma in the introduction.*

We added a Box to the Supplementary Information explaining the diversity metrics.

#66) l.322-324. *Long sentence a bit hard to follow. Please rephrase and please reword “the fact, that however”*

The text reads now in L369-372: “The finding highlights the fact that anthropogenic habitat types can still create heterogeneous communities. Natural habitat types have been mostly lost in the Anthropocene, causing ongoing biodiversity decline, but human-modified habitat mosaics can have a beneficial impact on community heterogeneity”.

#67) l.325. *A lot of “however” in the justification. Please consider rewording.*

L372-373: “On the other hand, at a global scale, this effect disappears, where net species diversity is declining.”

#68) l.339. *It is very clear in the methods. This is not consistently the case throughout the text, which might become clearer with revisions.*

Scales are now only used only as ‘local land-use’ and ‘regional landscape’.

#69) l.340-341. *Please consider rewording this sentence.*

The repetition was corrected, L387-390: “In each regional landscape, local land-use types were further determined based on the most dominant land-use type and vegetation on a 0.5 ha radius. Four local land-use was determined: forest (n=55), grassland (n=45), arable land (n=44) and settlement (n=35).”

#70) l.341. *on a resolution?*

Resolution referred to the remote sensing nature of site selection. To be more specific, we corrected this term, L387-389: “In each regional landscape, local land-use types were further determined based on the most dominant land-use type and vegetation using an area of 0.5 ha radius.”

#71) l.342. *I assume you are still referring to regions. It is important to define one term and consistently use it throughout the text, as the current wording makes it difficult to discern the scale you are discussing—whether It is regions, plots, or land use types. This issue recurs throughout the article, and maintaining clarity would greatly benefit from consistently using three main terms.*

We clarified our setup in L389-395: “Four local land-use was determined: forest (n=55), grassland (n=45), arable land (n=44) and settlement (n=35). In each regional landscape type, three study plots were set up, in a combination of three different local land-use types, out of the four possible land-use types. This resulted in 179 study plots belonging to four local land-use type, covering a ~ 1000 m elevational gradient, mean annual temperatures ranging from 5 to 10.3 °C, and precipitation from 550 and 1961 mm, distributed in the whole state of Bavaria, Germany³⁹”.

#72) L.346. *“for details see”?*

Corrected.

#73) l.352. *A variety of forest types are covered in your study, but this fact is not discussed anywhere in your article in regard with your result. Although you discuss a lot the variety of crops that may enhance heterogeneity in your arthropod’s community, you do not discuss why*

arthropods community are not that heterogeneous in forest? I may have the same remark for settlement plots that are taken in green areas, that may largely vary from one settlement to one another in term of plant species, exposure to the sun, temperature ... I think something interesting is missing in the discussion of your research.

That is true, there is not a high diversity of forest types in Bavaria. The sites were selected as following: L400-401: “Forest plots were in mostly managed forests, where broadleaf forest was preferred over coniferous ones”.

Some discussion on results from forest land-use type can be now found in L243-250:” The most natural local land-use type did not have the strongest distance-decay as expected, meaning that forests did not harbour largely different sets of species between locations. The arthropod communities with higher-than-expected homogeneity in forests are likely driven by current and historic Central European forest management. The mostly uniform management practices lead to even-aged stands with mostly closed canopies and lacking old-growth tree patches⁶³⁻⁶⁵. Consequently, even though forests had the highest α -diversity (meaning that this land-use type is the most locally diverse in arthropod species)⁶⁶ the created analogous habitats in forests lead to communities consisting of similar species between forest patches⁶⁷”.

And on settlements in L278-282: “Differences in practices are also present in settlements, but at a smaller scale: gardens in settlements have a huge range of grown plants and types of garden maintenance. This way, even though urban expansion is a major cause of habitat loss for many species^{85,86}, increasing attention is given to the beneficial impacts of dense habitat mosaics for species in settlements⁸⁷⁻⁸⁹.”

#74) L.380-381. Please reword not using mathematical abbreviation. Also, you are using two times the same reference to superior to 3 percent (>3%) but say “while” so I guess one should be inferior to.

L476-478: “BINs with less than 3% genetic distance were identified as BIN species, while those with more than 3% genetic distance served as “genetic morpho-species” in ecological assessments.”

#75) l.386-387. You say two times the same thing in two different sentences.

Corrected, thank you.

#76) l.389-390. *Reading this sentence, I do not understand which method has issues. "This method"*.

WE clarified the sentence in L486-489: "Using incidence data should also control for the fact that arthropod populations can be highly patchy in distribution both temporally and spatially, therefore abundance data could lead to species being over-, or underestimated in each sample community."

#77) l.392. *Can you please be clearer on the fact that abundant data may lead to underestimated species?*

This has been clarified and more details added in L485-499: "We used frequency instead of raw reads from the sequenced data, which has issues due to sequencing errors¹²⁸. Using incidence data should also control for the fact that arthropod populations can be highly patchy in distribution both temporally and spatially, therefore abundance data could lead to species being over-, or underestimated in each sample community. It has been shown that replicated incidence data support statistical approaches to diversity estimation that are just as powerful as corresponding abundance-based approaches^{129,130}. Incidence data can be substituted for raw abundance data when the true number of individuals is not available¹²⁹, such as data derived from metabarcoding. While our frequency approach cannot estimate species aggregations¹³¹, it does detect spatial and temporal community aspects better than abundance-based techniques¹²⁹. In the current study, we distinguish between infrequent, frequent, and highly frequent BIN species within each community of the four land-use types (see below in section "Rarity"). Nevertheless, it is important to note that frequency data represent the presence of species, not the exact number of species per sample, however this should not change the overall patterns of whole-of-community responses".

#78) l.396-397. *I do not understand here what are the fractions you are talking about?*

This was mentioned in the "sample collection" part of the methods, L423-426: "Samples were then split in two fractions, containing small and large species (or body parts) by sieving them through an 8mm sieve to control for differences in biomass and to increase the identification of rare and small species^{117,118}. "

#79) l.398. *In the method you start talking about sample and now you use assemblages. Maybe define that sample and assemblages are the same at the beginning and then only use sample?*

Methods have been updated and more details added. L515-517: “Measures of β -diversity (and distance-decay) is calculated by gamma-diversity divided by alpha-diversity, which means that this metric depends on between-sample differences and therefore dependent on sample coverage.”

#80) L.402-404. *Can you be clearer?*

L515-527: “Measures of β -diversity (and distance-decay) is calculated by gamma-diversity divided by alpha-diversity, which means that this metric depends on between-sample differences and therefore dependent on sample coverage. However, it is shown that most samples are incomplete, thus standardising sample coverage becomes very important ^{49,126}. 'Sample coverage' was originally developed by Alan Turing in his cryptographic work, which is defined as the proportion of the total number of individuals in the entire assemblage that belong to the detected species in a sample. Chao et al. ¹²⁷ extended Turing's concept of sample coverage to its incidence version to quantify sample completeness of replicated incidence data. This extension is based on the distribution of incidence frequencies obtained from multiple sampling units (such as DNA samples with identified BIN species). In our study, sample coverage may differ between local land-use types, therefore we used coverage-based indices calculated by the package *iNEXT.beta3D* ⁴⁷, which enables us to compare samples from different land-use types without any detection bias in our samples.”

#81) L.405. *Space missing when starting new sentence.*

Text has been updated.

#82) L.419. *Mistake in the bibliography citation?*

Corrected, thank you.

#83) L.423. *Maybe “however” is not the appropriate word here to justify your idea.*

“Although” might be better?

This has been corrected, L532-534: “Originally Hill numbers were developed for abundance data, this approach was extended to replicated incidence data”.

#84) 1.413-429. *On average, this is clear, but a formula might better illustrate what are those “q” value?*

There is no simple formula for Hill-numbers, and we did not want to detail the framework in our current study. Details can be found in Chao et al., 2014 “*Rarefaction and extrapolation with Hill numbers: a framework for sampling and estimation in species diversity studies*”. We included some details in L538-540: “For $q = 0$, the incidence-based Hill number reduces to species richness and thus is more sensitive to infrequent species; the measures with $q = 1$ and $q = 2$ can be interpreted as the effective number of frequent and highly frequent species in the community, respectively.”

#85) *In general, as your method is complex in term of duplication/steps, it might be a good idea to add a Figure (supplementary or in the main paper, as Figure 0 may be improved) so that everything is clear for the reader. It is hard to remember the scales, the index and so on. The steps of analysis are now included in the figure (was Figure 0, now Figure 1).*

#85) 1.436-437. *What were your criterion to decide this classification. It seems this is only based on your data and not on physiological value or other biological explanation. Could you please specify just with a small sentence in addition to the supplementary?*

The classes were based on the distribution of data. The dataset determined the body size classes, because if we set classes independent from the occurring size values, we would potentially lose information and introduce bias to our study. The sampling method has limitations, as mentioned in L415-421 “Nevertheless, this method reliably represents most groups of terrestrial arthropods¹¹⁴, including flying and flightless groups (also shown in Supplementary Fig.1): ground-foraging arthropods climb up on the net, and some small insects can be carried by the wind where the net intercepts this passive flight. Even though these traps are less effective in catching large flying insects with improved vision (such as dragonflies), they are still the most efficient trapping method for mass-collection of arthropods”.

We wanted to know the minimum and maximum body sizes based on real data, not a predetermine these values. We included this in L555-557: “Body size classes were established based on the distribution of the data, instead of a predetermined approach. The classes were determined based on the body size minimum and maximum values of the occurring BIN species.”

#86) 1.465. Please remove the extra point in the middle of the sentence.

Corrected, thank you.

#87) 1.492-493. Maybe remove “on their land”, it is a bit obvious.

Removed, thank you.

Additional remarks

#88) - Why do you have doi for some papers and not for others?

Citations are now plain text to avoid formatting issues with other devices when downloaded.

#89) - General remark for figure, maybe change $q=0$ for rare species in the figure instead of explaining what it is in the caption.

All figures include $q=0$, $q=1$, $q=2$ to avoid explanations within figures.

REVIEWER COMMENTS

Reviewer #1 (Remarks to the Author):

I'd like to commend the authors for the significant revision of the manuscript. As stated in my previous revision, this is an interesting and timely manuscript. Its also nice to see metabarcoding being used for consequential ecological questions. The manuscript has improved, especially relating to the details of the molecular method. However, I still have several issues relating to the molecular methodology and especially relating to my earlier comment on using detection frequencies as a proxy for abundances or rarity.

Reply: Thank you for the review. We included more clarification in the methods section to clarify two main points: 1) the issues around our species classification of infrequent, frequent and highly frequent categories. This classification is not *a priori* derived by global species distributions or conservation statuses. Instead, our data determines these categories based on their frequency within sampling units and within land-use categories, *a posteriori*. 2) more details were added to the DNA sequencing methods.

Responses contain more details below.

1) While the presentation of the molecular data has considerably improved, I'm still missing details here. The authors have provided a sophisticated statistical analysis and explained it in great detail. But the analysis will only make sense, if the underlying data are reliable. Readers need to be able to judge the reliability of the data based on the manuscript. I still can not fully do this right now. I'll explain what's still missing: A detailed treatment of the negative controls run along all samples is lacking. L464-466 "False positives were ... removed below 0.01% of total reads and by verifying OTU reliability against GenBank and BOLD .."

This is all, which is explained and a good first step. But this is not enough to remove false positives. You should use the negative control to inform about the level of background contamination. You say you have six controls per plate, but then never talk about them anymore. How were the negative controls used to inform data cleaning? Please explain in more detail. How did you deal with NUMTs? Is it possible that NUMTs inflate the recovered taxon diversity for groups with a high NUMT frequency?

Reply: We included more details in L471-489: "To minimize the impact of mitochondrial pseudogenes (NUMTs), we implemented multiple filtering steps, including de novo chimera detection and removal, length-based filtering, and taxonomic validation using BLAST against a custom BOLD database and GenBank, in combination with classification via the RDP

classifier. Taxonomic assignments were further refined through a consensus approach to minimize misclassification. Additionally, we implemented a stringent negative control-based filtering step: OTUs detected in any biological sample with read counts below the maximum observed in negative controls were excluded from further analysis. This step helps control for NUMTs that may arise from cross-contamination or environmental DNA artifacts. Given these multiple layers of quality control, we are confident that NUMTs have not significantly inflated the recovered taxonomic diversity in our study. “

However, if the reviewer has suggestions for additional NUMT-specific filtering approaches, we welcome further discussion.

You aimed at 100 k reads per samples, but what was the achieved coverage per sample in the end? How did you deal with uneven sample coverage between samples? Where all samples sequenced to exhaustion, or did you rarefy the data to some coverage level? Please show/explain this in the manuscript.

Reply: We clarified these details in the manuscript to enhance transparency in sequencing depth and data processing.

L480-488: “Our initial target sequencing depth was 100,000 reads per sample. In the final dataset, an average of 103,000 paired-end reads per sample was generated. After quality filtering, approximately 40,000 reads per sample remained, with ~30,000 reads retained after dereplication for downstream analysis. To address uneven sample coverage, we applied a normalization approach that ensured each sample was analysed in proportion to its retained sequencing output, avoiding artificial rarefaction. This method preserves as much biological information as possible while maintaining comparability across samples. Importantly, all samples were processed at full sequencing depth rather than being subsampled to a uniform coverage threshold. “

2) The other issue remaining is the use of frequency information to distinguish rare from common taxa. You are citing references 129-131 to justify your approach. I have no doubt that frequency data is very useful in ecological analysis. I even think using frequency of occurrence is probably is a good way of approximating abundance classes from DNA data. But I'm not sure, if this has been properly tested yet. References 129-131 are all references from the general ecology literature. None relates to metabarcoding data. Please cite metabarcoding literature, which suggest using frequencies as an abundance/rarity substitute.

Reply: We would like to clarify that in our approach, we did not treat incidence-based frequency as a substitute for species abundance. Instead, we used a model specifically formulated for incidence data. Although the original framework of Hill numbers and pertinent dissimilarity measures were developed based on species abundance/frequency data (Hill (1973) “*Diversity and evenness: a unifying notation and its consequences*” in *Ecology*, 54(2), 427-432), they were adapted for use with incidence or occurrence data (Colwell et al. (2012) “*Models and estimators linking individual-based and sample-based rarefaction, extrapolation and comparison of assemblages*” in *Journal of Plant Ecology* 5, 3-21). Under the adapted model for incidence data, only species detection/non-detection in any sampling unit (or in any of possible categories) is required; species abundances are not needed. Such a type of incidence data can be applied to any sampling as long as species detection/non-detection information can be collected in each sampling unit or category.

The model we use is not sensitive to the type of technique the data is derived from, therefore it is not a problem, that our data is produced by DNA-based techniques. Indeed, incidence data is often produced by either camera traps, audio recorders, field recording and any methods where an individual was not captured and identified. However, we included some literature (131 – 133) which specifically target DNA-based techniques when discussing incidence-based methods within the framework of Hill-numbers.

L530-534: “Incidence data can be used instead of raw abundance data when the true number of individuals is not available¹²⁹, such as data derived from metabarcoding¹³¹⁻¹³³. To handle the nature of incidence data, we use models specifically formulated for incidence data. In this model only species detection/non-detection in any sampling unit or category is required; species abundances are not needed.”

- 129 Colwell, R. K. et al. (2012) Models and estimators linking individual-based and sample-based rarefaction, extrapolation and comparison of assemblages. *Journal of Plant Ecology* 5, 3-21.
- 131 Buchner, D. et al. **Upscaling biodiversity monitoring: Metabarcoding estimates 31,846 insect species from Malaise traps across Germany**. *Molecular Ecology Resources* 25, e14023 (2025).
- 132 Alberdi, A. & Gilbert, M. T. P. **A guide to the application of Hill numbers to DNA-based diversity analyses**. *Molecular Ecology Resources* 19, 804-817, <https://doi.org/10.1111/1755-0998.13014> (2019).
- 133 Deagle, B. E. et al. **Counting with DNA in metabarcoding studies: How should we convert sequence reads to dietary data?** *Molecular Ecology* 28, 391-406 (2019).

You argue that by dividing each sample into two size classes you double your replication. I do not think this is true. Species which occur in both size classes are not frequent species, but species, which have a wide range of bodysize. This could for example be a species, which you find as adult and larva in the same trap. Species in only one size class are those which are only big or only small.

Reply: We thank the reviewer for this thoughtful comment. We have removed the word “replication” in our manuscript. Instead, we have explained that in our model based on incidence data, it is assumed that any species can be classified into one of the 16 possible categories (i.e., two size-class \times 8 sampling-time) and only species detection/non-detection information is recorded in each category. The incidence-based model developed in Colwell et al. (2012) for this case is approximately valid.

L548-554: “Frequency data ranged between 0 and 16: repeated temporal sample collection (n=8) multiplied by two sample fractions based on body size (‘big’ and ‘small’, described in sample collection). In our incidence-based model, any species could be classified into one of the 16 possible categories of two size-classes (‘big’ and ‘small’) multiplied by 8 sampling points in time. Body parts of big species (such as antenna) can be sieved into the ‘small’ fraction, and small species or body parts could end up in the ‘big’ fraction attached to big species (e.g. mites).”

Also, the issue still remains that using frequencies of occurrence across 8 time intervals does not necessarily indicate frequent species, but species with a temporally stable occurrence. So, your way of using frequency of occurrence across 16 subsamples across several months per site will not show common vs. rare species. Instead, it will show species with a wide size variation (e.g. different ontogenetic stages) and long-term temporal occurrence. Some flies, for example, can be hugely abundant at a site, but only for a few weeks. A species like this would be rated rare in your approach, because you only find it in one time window. Please explain how this is not affecting your inferences in the methods and discussion.

Reply: Our incidence-based approach will affect the inferences and discussion, and we do acknowledge that if species counts were used, the classification of rarity would be a more correct term and could change community compositions over Hill-numbers. However, we did not use species counts, given the metabarcoding data, and we believe that the broad patterns we discuss would be the same. However, this cannot be tested, because there is no count data available covering all-arthropod communities as accurately as the metabarcoding data derived from Malaise traps.

Our definitions of “infrequent”, “frequent” and “highly frequent” refer to the incidence-based frequencies among the 16 categories. That is, the three descriptions “infrequent”, “frequent” and “highly frequent” should refer to size × time categories. It is true, that our approach classifies species “infrequent”, which are only occurring for a short window in time (we don’t use the term “rare” or “common”). Given that we do not use IUCN or similar lists to classify the status of species, this should be a valid approach. Species which are only present for a short amount of time over the whole activity period should not be considered ‘frequent’ based on their abundance, and ignoring their spatial and temporal aggregations.

L132-139: “Rare and infrequent species often drive observed diversity patterns, therefore we accounted for changes in communities focusing on infrequent, frequent and highly frequent species separately⁵⁹. These categories differ from abundance or count-based classifications of species but reflect the incidence-based frequencies among the processed sample units (collection times and sample fractions, see Methods). Species data was processed with a deterministic approach, and frequency classes are identified based on frequencies within samples *a posteriori*, which does not reflect the geographic distribution or conservation status of a given species.”

L534-547: “While our frequency approach cannot estimate species aggregations¹³⁵, it does detect spatial and temporal community aspects better than abundance-based techniques¹³⁰. Following the example of Colwell et al. (2012), considering two species with the same number of individuals, but one species has a patchy distribution, and the other species occurs randomly. In this case, abundance-based techniques will be blind to association (and disassociation) and species aggregation. This topic is especially important in arthropod conservation. Optimal habitats are often patchy (for example veteran trees) and can have a high number of specific species’ individuals aggregated at a single tree. Even if there is a high local abundance of this species, it should be considered infrequent (or even “rare”), because it won’t occur at any other habitat type or at another tree. In this case, it is important to use incidence-based techniques. In the current study, we distinguish between infrequent, frequent, and highly frequent BIN species within each community of the four land-use types (see below in section “Frequency”).”

One way of finding out if the classification of rare and frequent species is correct, would be to explore the actual taxon lists. You could provide a table with species in the different rarity classes and explain, if the grouping makes sense. Afterall, the authors seem to know their

ecosystems very well. So, they should be able to show with their data that the classifications for species into rare frequent is actually true.

Reply: The classifications we use are not determined by *a priori*, they are determined by the models we use and are dependant of the sample-wide community. Therefore, this is not possible in a robust and reliable way using taxon lists and compare it to community-based classifications. Our models do not classify species into categories *per se*, they weigh species according to their incidence-frequencies within a sample, therefore there is no specific classification of species, which would be independent of the sample (taxon lists for example). Frequency classifications are made relative to the whole community (e.g. community of forest arthropods of plot 001 VS community of meadow arthropods of plot 002), therefore it is impossible to intercept species status (from IUCN lists for example) with our sample-dependent frequency values. The same species could be “infrequent” in one sample, but “highly frequent” in another.

L131-138: “Rare and infrequent species often drive observed diversity patterns, therefore we accounted for changes in communities focusing on infrequent, frequent and highly frequent species separately⁵⁹. These categories differ from abundance or count-based classifications of species but reflect the incidence-based frequencies among the processed sample units (collection times and sample fractions, see Methods). Species data was processed with a deterministic approach, and frequency classes are identified based on frequencies within samples *a posteriori*, which does not reflect the geographic distribution or conservation status of a given species.”

L592-599: “Species matrices based on their frequency were created using their overall frequency in the dataset. This variable is purely based on detection frequency within a community, and did not take their functional contribution or geographical range into account¹³⁹. This way, all frequency classifications are sample-dependent, as infrequent, frequent and highly frequent categories were assigned to species compared to others within the same sample. Therefore, the same species could be infrequent in one sample, but highly frequent in another one. All categories refer to the sample-scale frequency, i.e., local infrequent, local frequent and local highly frequent species within the sampled community.”

L411-412: “Samples were collected, and a limited number of samples were processed already by Uhler et al., 2021. However, for the current study, the full collection was processed.” So, the data was generated newly for this work, but the specimens were collected by Uhler et al. 2021 already? Then state it explicitly.

Reply: We are sorry for the confusion, the text states the exact steps now.

L431-434: “Samples were collected, and a limited number of samples were processed already by Uhler et al., 2021 using a slightly different approach in bioinformatics. However, for the current study, a new dataset was generated including the full dataset, processed with uniform bioinformatic techniques.”

Supplementary Figure.1. Please provide more detail in the caption. I can not follow what is exactly shown here, please explain better. 1a) average frequency per plot? Is this the number of OTUs found per plot for each taxon? Why would Dermaptera be present in so many OTUs at the plots? Isn't there only a handful of species for the group in Germany?

Reply: We thank the reviewer for noticing that the caption might be unclear. The frequency values do not reflect overall diversity of the orders, only the number of times species of this order was recovered from the sample. This means that one species of the order Dermaptera can be present many times within a given plot (between 0 -16 times max. of 179 study plots) or land-use category (for example between 0 – 16 times within 0 – 55 forest plots). The caption now reads as follows:

“Supplementary Figure.1. The graphs show the (a) average frequencies of recovered BIN species across taxonomic orders per study plot, i.e. possible 0 – 16 multiplied by 35-55 plots of various land-use category (forest=green, grassland=purple, arable land = golden, settlement=grey); and (b) the total frequencies of recovered BIN species across taxonomic orders on all study plots of LandKlif. Y-axis is plotted on a log-scale. The pie chart (c) shows the total reads of BIN species across 10 taxonomic orders with the highest number of reads.”

L604 Data availability: A link is missing here to the OTU tables and raw sequence data.

Reply: We are sorry for this. We uploaded the required datasets; however, we were notified that the link was only functional after several days, given that the DOI number was only changed internally after the original submission (likely that it was not changed to the reviewers). We upload the dataset again.

Reviewer #2 (Remarks to the Author):

Review Nat-COMM-24-35277A-Z

Reviewer Response Summary for Decker et al.

The study aims to investigate the role of land use in shaping beta diversity patterns of arthropods. The authors analyze arthropod communities based taxonomic diversity and on functional traits (body size and mobility) along a land-use intensity gradient within three different landscape types, employing distance decay analyses. The study stands out due to the extensive fieldwork conducted by the authors, who implemented large sampling techniques to collect a comprehensive dataset. Additionally, the research benefits from a robust laboratory pipeline, which allows for precise taxonomic identification and trait analysis. This combination of large-scale field sampling and cutting-edge genetic methodologies enhances the study's impact, providing valuable insights into the mechanisms driving community homogenization. Their hypothesis suggests that more natural habitats should exhibit greater community heterogeneity, while those embedded in highly human-modified landscapes should show increased homogenization. Focusing on arthropod communities, the study revealed unexpected patterns, the most homogenized communities were found in grasslands, whereas settlements and arable lands exhibited the highest diversity between locations. Beyond community homogenization, they also explored species traits within these communities. As anticipated, larger, and less mobile species exhibited more spatial heterogeneity than smaller, more mobile species. The title of the submitted article highlights the complexity of these findings.

The manuscript has benefited significantly from the reviewers' comments, and the authors have addressed these points effectively. They have provided additional justifications, analyses, and explanations that strengthen their study, adding depth to their findings. The aims of the study are now more explicitly stated in the introduction, improving clarity. The authors have now provided a more precise explanation of their expectations regarding species trait homogenization, particularly in relation to body size and mobility, with better contextualization within the existing literature. The discussion section is now more structured, making the arguments clearer and more cohesive. The inclusion of supplementary materials is particularly useful, offering valuable background on the study's vocabulary and methodology. The newly structured Figure 1 is greatly appreciated, as it aids in navigating the study's complex data and analytical framework.

My main remaining remark concerns the need to clarify and homogenize the terminology throughout the text. Although the authors have made a huge effort regarding the vocabulary and provided a terminology box, there remains some inconsistency in the use of key terms, particularly “distance decay,” “species homogenization,” and “community similarity loss.” While “distance decay” refers to changes in species composition along a spatial gradient, “community similarity loss” seems to convey a similar concept. To enhance clarity and consistency, the authors should decide on a single term or explicitly define their usage of these terms within the text. In the results section, “community similarity loss” is used, while the discussion section references both “distance decay” and “species homogenization.” The interchangeable use of these terms may lead to confusion. A clearer, more uniform application of terminology would help reinforce the study’s conclusions.

Reply: Thank you for the positive review. We read the text carefully and made some wording changes to improve clarity around distance-decay and community heterogeneity/homogeneity. We updated the graphs and changed the term from “community similarity loss” to “strength of distance-decay”.

L77-81: “(2) there is a physical limit to species dispersal distances. Steep distance-decay slopes (hereafter ‘strong distance-decay’) represent less similar (hereafter ‘heterogeneous’) communities, while flat slopes, or weak distance-decay indicate more similar (hereafter ‘homogeneous’) communities between locations.”

L117-121: “We expected stronger distance-decay, resulting in less homogenisation impact of the four local land-use types when situated within near-natural regional landscapes; while distance-decay is expected to be weaker, resulting in a more pronounced homogenisation effect in local land-use types when situated within agricultural and urban regional landscapes.”

L168:172: “Then, when focusing on frequent species ($q=1$), distance-decay within forests and managed grasslands did not differ, but it was still significantly weaker on grasslands than on arable lands and settlements. In communities focusing on highly frequent species ($q=2$), arable lands had the strongest distance-decay, not different from settlements, but significantly stronger than forests and managed grasslands (Fig 3a, Supp. I. Table 1).”

L207-223: “Body size did not significantly influence distance-decay relationships in most land-use types. Significant differences were only detected in communities focusing on highly frequent species ($q=2$) in forests, arable lands and settlements. In forests, small arthropods had a stronger distance-decay than medium arthropods. Arthropod body size did not impact distance-decay in grasslands. In arable lands, medium arthropods significantly showed the

weakest distance-decay. In settlements, large arthropods had a stronger distance-decay than small ones. (Fig. 5, Supplementary I. Table 3).

Arthropod communities with different mobility categories exhibited different distance-decay relationships. In forests, high-mobility species had weaker distance-decay than low- and intermediate-mobility species in communities focusing on infrequent species ($q=0$). In the case of communities focusing on frequent and highly frequent species ($q=1,2$), low-mobility species were only significantly different from intermediate-mobility species. In managed grasslands, low-mobility species had significantly stronger distance-decay ($q=0,1$), except in communities focusing on highly frequent species ($q=2$). In arable lands, high-mobility species significantly showed the weakest distance-decay ($q=0,1$), but this significance disappeared in communities focusing on highly frequent species ($q=2$). In settlements, distance-decay was not different among species with different mobility categories (Fig. 6, Supp. I. Table 4).”

Minor issues and line-by-line comments

l.20. Please correct to Leipzig.

Reply: Corrected, thank you.

l.56. When are you referring to your vocabulary box?

Reply: Thank you for noticing the lack of mentioning Box 1. We inserted this now.

L55-57: “This process can be measured via changes in β -diversity, describing community-scale patterns driven either by the environment, or stochastic factors^{7,8} (see Box 1 for used definitions).”

l.103 to l.105. Here you do not describes what are your “local land-use” and “regional landscapes”. Maybe you could briefly name them here and relate them to your Figure 1?

Reply: Thank you for noticing, we changed the text to make it clearer.

L102-109: “We studied distance-decay at two scales: local and regional. ‘Local land-use’ is considered the immediate surrounding of arthropod traps (0.5 ha), which impacts arthropod communities the most⁴³. ‘Regional landscape’ describes the broad area (5.8 km x 5.8 km quadrant), not only the realised habitat of an arthropod. First, we tested distance-decay patterns at the local land-use scale, because arthropod assemblage variations are best explained by local habitat characteristics⁴⁴. Then, we assessed the outcome of these local

subsets in three types of regional landscapes, which could modify local processes⁴⁵⁻⁴⁷ (Fig. 1).”

l.155. The term community similarity loss feels ambiguous—could it refer to species turnover or simply a reduction in species richness? To me, this phrase seems new and unexpected in the results section, as it was not introduced or defined earlier. I understand it relates to the concept of distance decay, where species composition changes along ecological gradients—communities farther apart in space share fewer species compared to those closer together. Additionally, while I recognize that you utilize various beta diversity measures to analyze changes in species composition from less frequent to more frequent species, I am not fully convinced that community similarity loss is the best term to describe these patterns. Wouldn't it be simpler and more intuitive to frame this as heterogeneity and homogeneity?

Alternatively, you could describe it as decay in community composition. For instance, a steep slope would indicate high decay, representing more heterogeneous communities across distance, while a gradual slope would reflect slower decay and thus more homogeneous communities along the same distance gradient.

I suggest clearly defining your preferred terminology in the introduction and consistently using it throughout the results and discussion sections. This will help readers easily follow your references to homogenization or its opposite. Currently, similarity loss feels harder to grasp conceptually.

Reply: Thank you for the comment. We rephrased the expectations in the introduction and used only distance-decay and community homogenisation/heterogenisation in the whole text. We have renamed the y-axes on the relevant figures (Fig.3, Fig. 6-7) to “Strength of distance-decay”, and modified the captions accordingly.

L110-116: “As all our habitats are at least moderately managed, we hypothesized a continuous decline of β -diversity^{5,9}, resulting in homogeneous communities over spatial distance - measured by distance-decay -, with increasing land-use intensity. Distance-decay is expected to be the strongest in forests, meaning that communities are the most heterogeneous between locations. Then, distance-decay is expected to weaken, meaning that communities become homogeneous between locations with increasing land-use intensity: in managed grasslands, arable lands, and settlements (for exact definitions, see Supp. Box 1.).”

Here, you seem to suggest that frequent-species communities are more heterogeneous across distances within forested habitats, which is similar to managed grasslands. However, in

grasslands, species communities across distances were more homogeneous compared to arable lands and settlements. Perhaps linking this observation to a specific figure or table would help clarify your point further?

Reply: We believe that the mentioned patterns are all shown on Figure 3.

L.161 to 170. Compared to the previous version, those results are better stated and described.

Reply: Thank you for the positive comment.

l.179. Figure 4 have greatly benefited from editorial work of the authors.

Reply: Thank you for the positive feedback.

l.195. What does this imply? Does it mean that the pattern observed at the taxonomic level (BIN species) is preserved when considering body size classes (except for xxx)? In other words, is the distance decay pattern consistent regardless of whether the species are large or small? I would recommend simplifying the phrasing here to make your point as clear as crystal.

Reply: Thank you for the comment, we included a rephrased interpretation to the Discussion. L332-334: “Distance-decay patterns were mostly not impacted by body size. This implies that patterns observed on taxonomic level are independent of the body size of specific species and preserved when body size as a trait determines community composition.”

l.338. “distance decay depending on mobility”. Please rephrase.

Reply: Thank you, we rephrased this. Now, it read as following:

L351-352: “In contrast to body size, distance-decay patterns of communities based on species mobility was affected by local land-use and species rarity.”

l.346: Could it be that species with low intrinsic mobility become passively mobile due to human presence? For instance, in settlements, species communities composed of both low-mobility and high-mobility species appear similarly heterogeneous. This could be explained by the lack of optimal habitats for highly mobile species to exhibit greater heterogeneity compared to other mobility categories, as you discuss. Alternatively, it could be that settlements facilitate passive transportation, increasing the mobility of low-mobility species.

Reply: We thank the reviewer for this insightful idea, we included this alternative.

L363-368: “Alternatively, species might be passively transported by human activities, which was shown among harvestmen (Opiliones) in an urban setting¹¹². This process could result in mobility-independent patterns of distance-decay, and low-, intermediate-, and high-mobility species could be equally likely to occur anywhere in settlements, ignoring the physical distance arthropods are able to travel.”

L.383. You could refer to your nice Figure 1 for your study design.

Reply: Thank you, Figure 1 is now referred to in the introduction (L109) and in the Methods section, statistical analysis (L564).

l.397. Is 3x30 correct?

Reply: Yes, the study used a 3 m x 30 m strip for sample collection.

Reviewer #2 (Remarks on code availability):

Thank you for taking the time to run our code and we are sorry for the inconvenience it caused. All the used datasets and corrected code is now available via the Figshare repository:

<https://figshare.com/s/fda6129fe1cbe0316341>

Errors reported:

Either the loaded final csv matrix is false or the code needs to be updated as follow:

l.14 - 20 - 24 - 28: object name error

Reply: the final datasets are now uploaded and tested again.

l.16 - 21 - 25 - 29: error in the number of column

Reply: the error was corrected.

l.57: needs to add install package for : library(snowfall)

Reply: ‘snowfall’ is available to install directly from the CRAN library (with R version 4.3.2). We included “install.packages(snowfall)” to the code.

l.66: where does the abline value comes from, I suggest you add a muted code corresponding to the calculation of this value as you did for Cmax_joint_f l.59.

Reply: Thank you for noticing. Ablines were used from calculating the median of the coverage values of all samples (‘DataInfobeta3D’ function) using the frequency matrices (for example “freq_forest”). Code is now included in the correct place. “SC” means sample coverage.

```
# check the actual sample coverage (SC) values to decide which one to use (use 0.8 at the end)
```

```
>SC_f=DataInfobeta3D (t(freq_forest))
```

```
>hist(SC_f $`SC(2n)`)
```

```
>median(SC_f$`SC(n)`)
```

```
>abline(v='value of median',col="red") # median of sample covers
```

```
>abline(v='value of joint Cmax',col="blue") # Cmax_joint value
```

l.82: needs to add install package for library(vegan)

Reply: 'vegan' is available to install directly from the CRAN library (with R version 4.3.2).

We included "install.packages('vegan')"

l.91-92: for now with the data provided the matrices does not contain "samples" value required to run the function. Please modify accordingly.

Reply: the function was corrected, thank you.

Impossible to run the rest of the code. Please verify your provided data and accordingly modify the code so it can be run.

Reply: the updated code was checked and ran without problems. We included more datasets to the supplementary material, because many output tables were manually modified (using Excel). These tables are now available, ready to download and use with the provided code via Figshare: <https://figshare.com/s/fda6129fe1cbe0316341>.

REVIEWER COMMENTS

Reviewer #2 (Remarks to the Author):

Review Nat-COMM-24-35277B

Reviewer Response Summary for Decker et al.

This study investigates how land use intensity drive arthropod diversity loss by examining both taxonomic and functional traits (body size and mobility) along a land-use intensity gradient across four land-use types and at two scales. Using extensive field sampling and advanced lab techniques, the researchers identify patterns of community homogenization. Contrary to expectations, the most homogenized communities were found in grasslands, while settlements and arable lands showed higher diversity between sites. Trait analysis revealed that larger, less mobile species had greater spatial heterogeneity.

Following reviewer feedback, the manuscript was significantly improved: the study's aims were clarified, trait-based expectations better contextualized, and the discussion made more cohesive. I value the authors' effort in ensuring terminological consistency across the text. Aside from the brief remark below, I have no further comments to add.

Thank you for taking the time to read our manuscript again and thank you very much for the positive review!

Minor issues and line-by-line comments

l.93. Can you here already provide what are your four land-use types?

We added the land-use types in the text in L 92-93: "Here, we compared four land-use types of Central Europe along a gradient of land-use intensity in forests, managed grasslands, arable lands, and settlements at the scale of a large federal state - a scale most relevant for political decisions."

Reviewer #3 (Remarks to the Author):

The manuscript entitled “Distance decay reveals contrasting effects of land-use types on arthropod community homogenisation” addresses the highly relevant and timely topic of biodiversity homogenisation driven by land-use change, based on a sophisticated and extensive dataset. The authors present a comprehensive study that contributes valuable insights to the field. I have carefully reviewed both the manuscript and the authors’ responses to previous reviewer comments. While many aspects have been adequately addressed, several methodological and structural issues remain and require further clarification or revision before the manuscript can be accepted.

Thank you for the thorough review and the positive comment.

Specific Comments:

- *Line 449: Please provide the concentration of Proteinase K used in the extraction protocol to ensure full reproducibility.*

We included the information in L446-448: “DNA extraction from all samples was conducted by incubating them in a 90:10 solution of animal lysis buffer (buffer ATL, Qiagen DNEasy tissue kit, Qiagen, Hilden, Germany) with 10% proteinase K following the standard Qiagen DNeasy protocol.”

- *Lines 466–484: This section describing quality-filtering of the sequence data should be moved to the “Bioinformatics and Taxonomic Classification” section. Quality control and filtering are part of post-sequencing data processing and should logically follow the raw-data processing description.*

Thank you for the comment, these sections were moved.

- *Line 486: Please provide the citation for the USEARCH software.*

Thank you, we inserted the following citation:

126: Edgar, R. C. Search and clustering orders of magnitude faster than BLAST. *Bioinformatics* 26, 2460-2461, doi:10.1093/bioinformatics/btq461 (2010).

- *Line 488: Cutadapt should be cited, and the version number used in this study should be included.*

Thank you, we inserted the following citation:

127: Martin, M. Cutadapt removes adapter sequences from high-throughput sequencing reads. *2011* 17, 3, doi:10.14806/ej.17.1.200 (2011).

- *Line 491: Please cite VSEARCH appropriately.*

Thank you, we inserted the following citation:

128: Rognes, T., Flouri, T., Nichols, B., Quince, C. & Mahé, F. VSEARCH: a versatile open source tool for metagenomics. PeerJ 4, e2584, doi: <https://doi.org/10.14806/ej.17.1.200> (2016).

- *Line 492: Why was filtering only based on minimum read length? It is standard practice, especially for COI metabarcoding, to apply both minimum and maximum length thresholds to remove nonspecific or chimeric fragments. Please justify this deviation.*

We acknowledge that applying both minimum and maximum length thresholds is recommended for COI metabarcoding. In our early pipeline, only a minimum threshold of 300 bp was used. However, given the consistent amplicon length (313 bp) and subsequent quality control steps (chimera filtering, taxonomic validation, and abundance-based filtering), we believe the lack of a maximum length filter had minimal impact on the final OTU set.

We included the following text in L478-480: “Initial quality filtering was implemented with `--fastq_filter` (parameters: `--fastq_maxee 1, --minlen 300`). Filtering to a minimum length was applied given the consistent amplicon length (313 bp) and subsequent quality control steps.”

- *Pre-clustering rationale: The manuscript states that pre-clustering was performed prior to chimera filtering. However, de novo chimera filtering tools are typically efficient and fast. Please clarify the purpose and benefit of this step, as it is not clearly justified and may introduce unnecessary complexity.*

We acknowledge that performing chimera detection on the full dereplicated dataset is standard. However, due to the computational demands of *de novo* chimera detection using VSEARCH's `--uchime_denovo` at the time of our original analysis, we implemented a 98% identity pre-clustering step. This approach, inspired by discussions in the VSEARCH community, aimed to reduce the dataset size and, consequently, the computational load by focusing chimera detection on cluster centroids. While this method may slightly reduce sensitivity to rare chimeras, subsequent quality control steps, including abundance filtering and taxonomic validation, help mitigate potential artifacts. This strategy was essential to feasibly process our large dataset with the computational resources available at the time.

We included this in the text, L484-489: “Following a 98% identity pre-clustering, chimera filtering was done using the centroids algorithm and VSEARCH’s --uchime_denovo utility. This step reduced the size of the dataset and therefore the computational load by focusing chimera detection on cluster centroids. While this method may slightly reduce sensitivity to rare chimeras, subsequent quality control steps, including abundance filtering and taxonomic validation, help mitigate potential artifacts.”

• *Line 498: The statement about using “BLAST” against the BOLD database is inaccurate. BOLD does not support direct BLAST-based queries in the same way as GenBank. Please clarify the exact procedure:*

o When were the sequence similarity searches performed (I saw years in the read table)?

o Which BLAST algorithm was used (e.g., blastn, megablast)?

o Was a local database used or an online search?

o For GenBank, which database subset was queried?

o How was barcode identification against BOLD conducted - was this done via BOLDigger or directly via the website interface?

Thank you for the comment. This was done using two separate BLAST databases (NCBI nucleotide (nt) database and a custom-formatted BLAST database constructed from COI records retrieved via the BOLD public API <http://www.boldsystems.org/index.php/resources/api>). BIN assignments were based on the best-hit from this custom BOLD database. This allowed us to bypass BOLD’s web interface, which does not support direct BLAST queries, and maintain full control over query parameters and annotation consistency.

We included some clarification in the text in L504-515:” BLAST searches were conducted locally using the blastn utility from NCBI BLAST+ (<ftp://ftp.ncbi.nlm.nih.gov/blast/executables/blast+/>), with the following parameters: -task megablast -max_target_seqs 1 -max_hsps 1 -evalue 10 -word_size 28 -outfmt "6 sacc pident salltitles qseqid length". For each OTU query, only the top match by percentage identity was retained. Searches were run against two separate BLAST databases: 1) The NCBI nucleotide (nt) database, downloaded from <ftp://ftp.ncbi.nlm.nih.gov/blast/db/> (download date recorded in the result tables); and 2) custom-formatted BLAST database constructed from COI records retrieved via the

BOLD public API (<http://www.boldsystems.org/index.php/resources/api>), parsed from TSV exports. Each BOLD record was annotated with a BIN (where available), BIN assignment was based on the best-hit from this custom BOLD database. This allowed us to maintain full control over query parameters and annotation consistency.”

• *Line 498 (read threshold): Please specify whether the 0.01% abundance threshold was applied globally across the entire dataset or per sample. This influences downstream diversity measures and interpretation.*

Thank you for the comment, we inserted more details about this step in L515-521: “The 0.01% abundance threshold was applied per sample, prior to taxonomic assignment. This conservative filtering step was used to minimize noise from index hopping, tag-jumping, and stochastic PCR or sequencing errors, which could introduce spurious low-frequency reads¹²⁹⁻¹³¹. Per-sample filtering ensures that true low-abundance OTUs in complex samples are retained if consistently present, while sample-specific artifacts are excluded. This approach is particularly relevant for large, heterogeneous datasets where global thresholds risk discarding real but rare taxa^{132,133}.”

129: Esling, P., Lejzerowicz, F., & Pawlowski, J. (2015). Accurate multiplexing and filtering for high-throughput amplicon-sequencing. *Nucleic Acids Research*, 43(5), 2513–2524. <https://doi.org/10.1093/nar/gkv107>

130: Schnell, I. B., Bohmann, K., & Gilbert, M. T. P. (2015). Tag jumps illuminated – reducing sequence-to-sample misidentifications in metabarcoding studies. *Molecular Ecology Resources*, 15(6), 1289–1303. <https://doi.org/10.1111/1755-0998.12402>

131: Zepeda-Mendoza, M. L., Bohmann, K., Carmona Baez, A., & Gilbert, M. T. P. (2016). DAME: a toolkit for the initial processing of datasets with PCR replicates of double-tagged amplicons for DNA metabarcoding analyses. *BMC Research Notes*, 9(1), 255. <https://doi.org/10.1186/s13104-016-2064-9>

132: Leray, M., & Knowlton, N. (2017). Random sampling causes the low reproducibility of rare eukaryotic OTUs in Illumina COI metabarcoding. *PeerJ*, 5, e3006. <https://doi.org/10.7717/peerj.3006>

133: Deagle, B. E., *et al.* (2018). Counting with DNA in metabarcoding studies: How should we convert sequence reads to dietary data? *Molecular Ecology*, 28(2), 391–406. <https://doi.org/10.1111/mec.14734>

- *Line 500: The rationale for using the RDP's Bayesian classifier after already conducting similarity searches against BOLD and GenBank is unclear.*
 - o *What was the added value of this second classification step?*
 - o *Which reference database was used with the RDP classifier?*
 - o *Overall, the taxonomic assignment workflow appears unnecessarily complex. Many studies, including comparable Malaise trap datasets, rely on a single step against BOLD. Please justify your multi-step approach and elaborate on its benefits over standard protocols.*

We utilized the RDP classifier (Wang *et al.*, 2007), trained on the COI reference set developed by Porter and Hajibabaei (2018), to complement our BLAST-based taxonomic assignments.

We included details in L522-529: “Taxonomic classification employed RDP’s Bayesian classifier¹³⁴ trained on the COI reference dataset¹³⁵ to complement our BLAST-based taxonomic assignments. The RDP classifier provides bootstrap support values for each taxonomic rank, offering a probabilistic measure of assignment confidence. By integrating RDP classifier outputs with BLAST results from GenBank and BOLD, we employed a least common ancestor (LCA) consensus approach. This method ensures conservative and robust taxonomic assignments, especially in cases involving degraded sequences or incomplete reference databases. The resulting taxonomic summaries were visualized through KronaTools v1.311.”

134: Wang, Q., Garrity, G.M., Tiedje, J.M., & Cole, J.R. (2007). Naive Bayesian classifier for rapid assignment of rRNA sequences into the new bacterial taxonomy. *Applied and Environmental Microbiology*, 73(16), 5261–5267. <https://doi.org/10.1128/AEM.00062-07>

135: Porter, T.M., & Hajibabaei, M. (2018). Automated high throughput animal COI metabarcoding classification. *Scientific Reports*, 8, 4226. <https://doi.org/10.1038/s41598-018-22505-4>

• *Line 502: BIN assignment is only possible via BOLD systems. Please explain how BINs were assigned in your workflow. Was a direct BOLD API or export used?*

This approach enabled consistent BIN-level assignment without relying on the BOLD web interface.

We included more details in L532-537: “BIN assignments were derived from BLAST results against a custom BOLD reference database built locally. COI sequences were downloaded via the BOLD public API in TSV format and used to construct a local BLAST database, in which most records included BIN designations in the fasta headers. After performing `blastn` searches, OTUs were assigned to BINs based on the top-hit match from this annotated BOLD database.”

• *Line 506: The term “dark taxa” must be clearly defined, contextualised, and supported by a reference. This term is not self-explanatory to all readers.*

We defined the term “dark taxa” in L539-544: “We use the term “dark taxa” to refer to Barcode Index Numbers (BINs) or molecular OTUs that lack formal Linnaean taxonomic names due to gaps in reference libraries. This usage follows established literature where “dark taxa” denote unclassified or provisionally assigned taxa in molecular datasets^{138,139}. These are common in metabarcoding of underrepresented groups such as Diptera and Hymenoptera, where many BINs remain unlinked to described species.”

138: Pentinsaari, M., Ratnasingham, S., Miller, S. E., & Hebert, P. D. N. (2020). BOLD and GenBank revisited—do identification errors arise in the lab or in the sequence libraries? *PLoS ONE*, 15(5), e0231814.

<https://doi.org/10.1371/journal.pone.0231814>

139: Hartop, E., Lee, L., Srivathsan, A., *et al.* (2024). Resolving biology’s dark matter: species richness, spatiotemporal distribution, and community composition of a dark taxon. *BMC Biology*, 22, 215. <https://doi.org/10.1186/s12915-024-02010-z>

• *Müller et al. (2023): The methodology for BIN alignment remains unclear. How were sequences aligned to BINs? Was a local copy of the BIN dataset downloaded and used for alignment? This step needs to be described in greater technical detail, especially if the method deviates from standard BOLD procedures.*

We added details in L546-554: “BIN alignment followed the protocol outlined in Müller *et al.*²². OTU sequences were aligned to the closest BINs in our custom BOLD reference database, which was locally constructed from BOLD COI data retrieved via the public API. For each OTU, we identified the top `blastn` hit with a BIN assignment and calculated the pairwise genetic distance. Sequences showing <3% uncorrected p-distance to a reference BIN were assigned as part of that BIN species; those >3% were considered “genetic morpho-species.” All alignment and distance calculations were performed locally using standard BLAST and sequence comparison tools, enabling consistency across the dataset and alignment to regional BINs relevant to our sampling scope.”

- *Line 520: Please revise the sentence to reflect that raw reads may be influenced not only by sequencing errors, but also by PCR errors, and primer bias.*

We agree that raw read abundances in metabarcoding are influenced by multiple biases, including sequencing errors, PCR amplification variability, and primer-template mismatches. These factors can lead to nonlinear relationships between true biomass and recovered, justifying our use of detection frequency as a more stable metric for ecological analysis.

The revised sentence can be read in L559-562:” We used detection frequency instead of raw read abundance data for a stable metric for ecological analysis. Read abundances can be influenced by multiple biases, including sequencing errors, PCR amplification variability, and primer-template mismatches^{140,141}.”

Additional reference added (141): Piñol, J., Mir, G., Gomez-Polo, P., & Agustí, N. (2015). Universal and blocking primer mismatches limit the use of high-throughput DNA sequencing for the quantitative metabarcoding of arthropods. *Molecular Ecology Resources*, 15(4), 819–830. <https://doi.org/10.1111/1755-0998.12355>

- *Figure 4: It appears that the diversity indices are mislabeled. Probably the plots were ment to be labeled as follows: $q = 0$; $q = 1$; $q = 2$.*

Thank you for noticing this, the figure was corrected.

Conclusion:

The authors have successfully addressed the majority of earlier reviewer comments and

the study is, in principle, suitable for publication in Nature Communications. However, several important issues remain, particularly in the Methods section, regarding clarity, reproducibility, and methodological justification. Once these remaining concerns are adequately resolved, I would support the publication of this manuscript.

REVIEWERS' COMMENTS

Reviewer #3 (Remarks to the Author):

The authors have adequately addressed all of my previous comments, and I have no further concerns. I can recommend the manuscript for publication in its current form.

Thank you very much.